# Action of CMG with strand-specific DNA blocks supports an internal unwinding mode for the eukaryotic replicative helicase

Lance Langston, Mike O'Donnell*

Howard Hughes Medical Institute, The Rockefeller University, New York City, United States

**Abstract** Replicative helicases are ring-shaped hexamers that encircle DNA for duplex unwinding. The currently accepted view of hexameric helicase function is by steric exclusion, where the helicase encircles one DNA strand and excludes the other, acting as a wedge with an external DNA unwinding point during translocation. Accordingly, strand-specific blocks only affect these helicases when placed on the tracking strand, not the excluded strand. We examined the effect of blocks on the eukaryotic CMG and, contrary to expectations, blocks on either strand inhibit CMG unwinding. A recent cryoEM structure of yeast CMG shows that duplex DNA enters the helicase and unwinding occurs in the central channel. The results of this report inform important aspects of the structure, and we propose that CMG functions by a modified steric exclusion process in which both strands enter the helicase and the duplex unwinding point is internal, followed by exclusion of the non-tracking strand.

*For correspondence: odonnel@mail.rockefeller.edu

**Competing interests:** The authors declare that no competing interests exist.

## Introduction

Cellular DNA replication in all domains of life employs a common set of proteins including RNA primase, DNA polymerases, a replicative helicase, a DNA sliding clamp and a clamp loader. In all cell types, the helicase is based on a hexameric ring that surrounds and moves along DNA using the power of NTP hydrolysis to bring about separation of the two parental strands so that each can serve as a template for a new daughter duplex (*Enemark and Joshua-Tor, 2008*; *Lyubimov et al., 2011*; *Nandakumar and Patel, 2013*). In many cases, the helicase also acts as a platform for binding and recruitment of other proteins necessary for DNA replication, in particular the primase which is required for periodic initiation of Okazaki fragments on the lagging strand (*Benkovic et al., 2001*; *Georgescu et al., 2015*). All hexameric helicases characterized to date bind NTP at the interface between adjacent subunits (*Enemark and Joshua-Tor, 2008*; *Lyubimov et al., 2011*; *Nandakumar and Patel, 2013*).

The replicative helicase of eukaryotes is based on this same arrangement of a core hexameric ring comprised of six related Mcm subunits, but helicase activity requires five additional proteins including Cdc45 and the heterotetrameric GINS complex (Psf1-3 and Sld5). The Michael Botchan lab coined the term CMG (Cdc45, Mcm2-7, and GINS) and demonstrated helicase activity for both the native complex (isolated from Drosophila embryos) and the recombinant 11 subunit enzyme (*Ilves et al., 2010*; *Moyer et al., 2006*). CMG is formed from its subcomponents in a highly regulated manner starting with the loading of two head-to-head Mcm2-7 hexamers at origins of DNA replication in a process mediated by the Orc1-6, Cdc6 and Cdt1 proteins (*Boos et al., 2012*). Cdc45 and GINS are chaperoned into the CMG complex during S phase by several additional proteins and

kinases (*Bell and Labib, 2016*). The structure of CMG, determined by negative stain EM 3D single-particle reconstruction, showed a central channel formed by the Mcm2-7 ring for encircling and tracking on the leading strand but also observed a second channel formed by the accessory factors at the side of the Mcm2-7 ring (*Costa et al., 2011*). The exact means by which the additional subunits contributed to helicase activity was unclear, but it was suggested they might encircle the lagging strand and help to partition the two parental strands (*Costa et al., 2011*).

Sequence alignments of hexameric helicases have defined four superfamilies, SF3-6 (*Singleton et al., 2007*). Two superfamilies are bacterial/phage (SF4, SF5) and their motors are built on the RecA motif; the two other superfamilies are eukaryotic/viral (SF3, SF6) and their motors are based on the AAA+ motif. Visualization of replicative hexameric helicases reveals two stacked rings due to the bi-lobed architecture of the individual motor subunits with distinct N- and C-domain tiers. In all cases, the C-terminal domain contains the ATPase sites. While the bacterial helicases translocate on ssDNA 5′−3′, the eukaryotic helicases translocate 3′−5′. Studies comparing co-crystal structures of the eukaryotic Bovine Papilloma Virus (BPV) E1 helicase (SF3)-ssDNA complex (*Enemark and Joshua-Tor, 2006*) with the *E. coli* Rho factor (SF5)-ssDNA complex (*Thomsen and Berger, 2009*) indicate that they both bind ssDNA the same way in their motor domains (*Thomsen and Berger, 2009*). The similarity in DNA direction through the motors also holds for *E. coli* DnaB (SF4)-ssDNA (*Itsathitphaisarn et al., 2012*) and the Mcm2-7 within eukaryotic CMG helicase (Cdc45/Mcm2-7/GINS) (SF6) bound to a forked junction (*Figure 1A*) (*Georgescu et al., 2017*). Given that the CTD of all hexameric helicases contain the motors, and taking into account, their opposite directions of translocation, bacterial hexameric helicases track on the lagging strand with the C-tier ahead of the N-tier, and the eukaryotic helicases track on the leading strand with the N-tier leading the way.

There are two major proposals for how hexameric helicases function during unwinding as illustrated in *Figure 1B* (*Li et al., 2003*; *Slaymaker and Chen, 2012*). The 'steric exclusion' model (*Figure 1B*, left) posits that the helicase encircles the tracking strand while the other strand is completely excluded from the interior of the helicase, thus acting as a moving wedge with the unwinding point external to the central channel. The alternative is the 'side channel extrusion' model (*Figure 1B*, middle), in which the helicase encircles both strands of dsDNA and the unwinding point is inside the central channel, with one strand being extruded out a side channel, usually proposed to be at a subunit interface between the N-tier and C-tier. There is a growing consensus that all

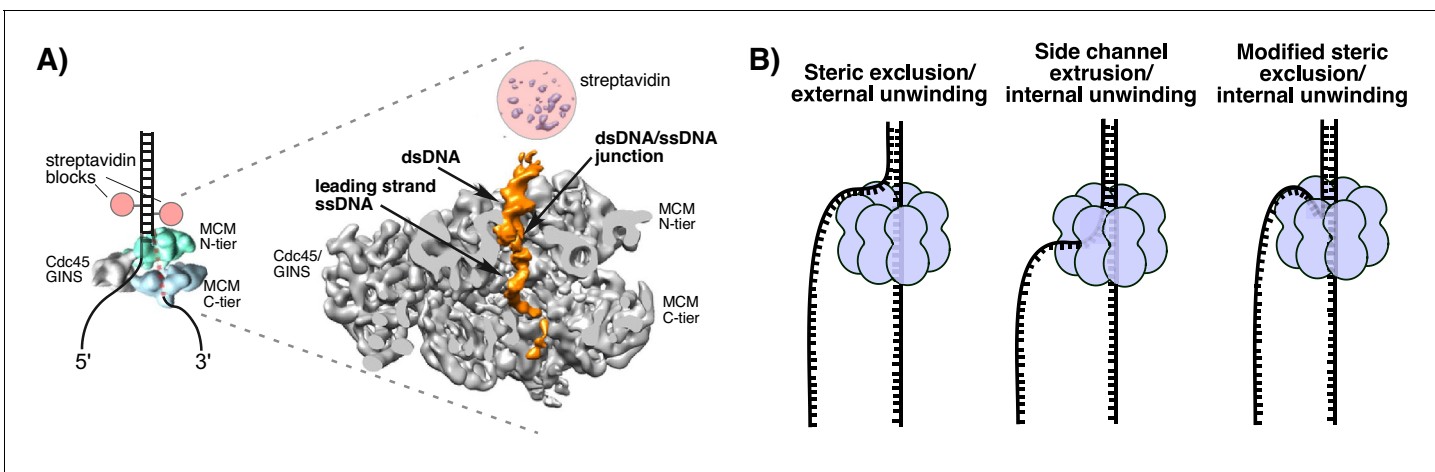

**Figure 1.** Structure of *S. cerevisiae* CMG at a replication fork and models of hexameric helicase unwinding. (A) CryoEM single particle 3D reconstruction of active CMG that was captured by streptavidin blocks (left diagram); the CMG is a surface rendering. The right panel is the CMG-forked DNA complex as a cut-open surface rendering. Adapted from *Figure 6* of *Georgescu et al. (2017)*. (B) Models of hexameric helicase function. Left: classic steric exclusion in which the helicase encircles only one strand, excluding the other and the unwinding point is external to the helicase. Middle: classic side channel extrusion model with duplex DNA entering the channel and the DNA split point is internal, with one strand extruded out a side channel. Right: Proposed modified steric exclusion model with duplex entering the channel and an internal unwinding point, followed by exclusion of the non-tracking strand.

hexameric helicases function by the steric exclusion model, although there are no crystal structures of a hexameric helicase at a replication fork. A common biochemical assay that distinguishes the two models is the use of strand-specific blocks (*Fu et al., 2011*; *Hacker and Johnson, 1997*; *Kaplan, 2000*; *Kaplan et al., 2003*). For example, the *E. coli* DnaB helicase is not inhibited by a block on the non-tracking strand (leading) but is strongly inhibited by a block on the tracking strand (lagging) (*Kaplan, 2000*). By this criterion, DnaB acts by steric exclusion because if the leading strand passed into the central channel and out through a side channel, a block on the non-tracking leading strand would have been inhibitory.

Using strand-specific blocks in the Xenopus extract system, CMG was also determined to act by steric exclusion (*Fu et al., 2011*). Replication forks in the *Xenopus* extract were strongly inhibited by streptavidin blocks on the tracking strand (leading) but were only transiently inhibited (partial inhibition at the first 10 min time point) by streptavidin blocks on the non-tracking (lagging) strand. The same study examined Dig-antiDig blocks in single-molecule studies and observed only 20–26% inhibition by the lagging strand block compared to 93–100% for the leading strand block (*Fu et al., 2011*). Hence, the results argued for steric exclusion along with some type of minor slow down caused by blocks on the lagging strand. To explain the slow down by the lagging strand block, it was suggested that lagging strand wrapping around the outside of CMG may be disrupted by the block and somehow slow the helicase. Lagging strand wrapping around an archaeal MCM had already been demonstrated (*Graham et al., 2011*) and DNA–protein cross-linking studies of *Drosophila* CMG supported lagging strand wrapping around the outside of CMG (*Petojevic et al., 2015*). The *Drosophila* studies also indicate that the leading strand can enter the second channel formed by the accessory factors at the side of the Mcm2-7 ring through opening of the gate in the Mcm2/5 subunits and that the Cdc45 subunit captures the leading strand, keeping it from exiting the interior of CMG. Alternatively, it was proposed that the lagging strand might pass through this second channel to achieve separation of the duplex (*Costa et al., 2011*), but this idea was not supported in the later study (*Petojevic et al., 2015*). High-resolution studies (3.7 Å) of the N-terminal face of Saccharomyces cerevisiae CMG show that the second channel is completely filled-in by the protein side chains (*Yuan et al., 2016*). Hence, in the yeast CMG, there is no room for the lagging strand to fit through. Moreover, the crystal structure of human Cdc45 reveals that the site proposed to bind DNA based on homology to RecJ nuclease is completely occluded (*Simon et al., 2016*).

These studies have led to the conclusion that CMG operates by steric exclusion like other hexameric helicases, but definitive insights into the mechanism of this essential helicase are lacking. In this report, we set out to determine how isolated *S. cerevisiae* CMG functions when presented with blocks on either the leading or lagging strand and compare the results to the complete replisome in the *Xenopus* system. In addition, about the time the current study was concluded, we succeeded in obtaining a 6.2 Å resolution cryoEM single-particle 3D reconstruction of an active CMG captured at a replication fork by a dual biotin-streptavidin block (*Figure 1A*) (*Georgescu et al., 2017*). Interestingly, CMG encircles a short region of dsDNA in the N-tier ring, and the unwinding point is inside the central channel where protruding loops from the OB fold subdomain of Mcm's lining the channel might facilitate unwinding; the C-tier motor ring is behind the N-tier ring. These striking features are unique to CMG thus far and run contrary to the steric exclusion model in which the helicase encircles only ssDNA and the unwinding point is external to the channel. The dsDNA does not sink far into the CMG helicase before being unwound and is surrounded by the zinc fingers at the N-terminal region but does not get entirely past the OB folds in the N-tier of the central channel. The dsDNA appears to be tightly held and rigid because it is well positioned in the structure at a 28° angle to the central channel and it seems likely that CMG contacts both strands to position DNA in such a rigid fashion. The unwound leading strand template ssDNA proceeds down the central channel into the C-tier ring, while the lagging strand template is not visualized in the CMG-forked DNA structure indicating either multiple locations or extensive mobility of this strand (*Georgescu et al., 2017*). Although dsDNA entry and internal unwinding are features of the side channel extrusion model (*Figure 1B* middle), assays using strand-specific blocks might still produce results consistent with the steric exclusion model if the lagging strand comes back out the central channel and thereafter remains outside CMG, as suggested in the structural study. We refer to this as a 'modified steric exclusion' model with dsDNA entry and internal unwinding followed by exclusion of the lagging strand back out of the central channel (*Figure 1B*, right). One advantage of this arrangement is that the Pol α-primase binds CMG through a Ctf4 bridge at the N-tier face (*Sun et al., 2015*). Thus, the

most efficient path of the lagging strand template would be to thread back out the top of the central channel to contact the Pol α-primase at the top (N-tier) of CMG. Similarly, the Pol ε leading polymerase binds directly to CMG on the C-tier face and thus is well positioned to copy the leading strand template as it emerges from the bottom of the central channel (*Asturias et al., 2006*; *Georgescu et al., 2017*; *Sun et al., 2015*).

To investigate the mechanism of CMG translocation in a defined system, we used purified CMG from *S. cerevisiae* along with a series of model forked DNA substrates to determine if unwinding is impeded by blocks on the leading and lagging strand templates. We find that biotin-streptavidin blocks on either strand strongly inhibit CMG unwinding and therefore support a model in which both strands of dsDNA enter the central channel of CMG, as visualized in the EM structure (*Georgescu et al., 2017*). The results were unexpected, as was the CMG-forked DNA structure, and the blocking data inform the functionality of the CMG-forked DNA structure. If CMG were to encircle only the leading strand, as posited by the classic steric exclusion model, then lagging strand blocks should not affect CMG activity. But if CMG normally binds dsDNA at the entry point to facilitate unwinding, as the cryoEM structure reveals, then blocks on the lagging (excluded) strand template should inhibit helicase activity. If the dsDNA connection is not very tight, and if the helicase is active without needing internal residues or dsDNA binding, the lagging blocks might show little or no inhibition of helicase activity. Indeed, *Xenopus* extract studies indicated a temporary slowdown by dual streptavidin blocks on the lagging strand, and only 20–26% of replication forks were halted when a Dig-Antidig antibody block was encountered on the lagging strand (*Fu et al., 2011*). However, in an extract, many other proteins could participate to reduce the slowdown. Thus, study of isolated pure CMG is needed to assess this unique feature of dsDNA entry into CMG helicase.

## Results

### CMG translocates over flush duplex DNA without unwinding

To investigate the mechanism of unwinding by CMG, we purified the 11-subunit complex as previously described (*Georgescu et al., 2014*; *Langston et al., 2014*) and examined its unwinding activity with radiolabeled DNA substrates representing the structures found at replication forks (schematics in *Figure 2—figure supplement 1* and *Figure 3—figure supplement 1*). Two recent cryoEM structures of *D. melanogaster* CMG and *S. cerevisiae* CMG were solved in the presence of a forked DNA, but only ssDNA was observed going through the entire central channel (*Abid Ali et al., 2016*; *Georgescu et al., 2017*). The central channel of the *D. melanogaster* CMG appeared too constrictive to bind dsDNA while the *S. cerevisiae* CMG-ssDNA had a central channel that could potentially accommodate dsDNA. Interestingly, an earlier structure of apo *S. cerevisiae* CMG without DNA showed a winged helix domain (WHD) projecting into the central channel that would have prevented dsDNA entry, but the WHD domain moved out of the channel in the CMG-ssDNA structure (*Georgescu et al., 2017*; *Yuan et al., 2016*). CMG is formed in vivo by the loading of Mcm2-7 onto duplex DNA and many hexameric helicases have been shown to translocate across a flush (non-tailed) duplex DNA without unwinding (*Kang et al., 2012*; *Kaplan, 2000*). Presumably, initiation of the unwinding reaction requires the helicase to encounter an impediment, like a tailed duplex. To determine whether duplex DNA can pass through the entire central channel of yeast CMG, we used a forked DNA with a flush duplex between the 3′ ssDNA loading site and the forked junction. Thus, CMG will need to traverse the flush duplex to reach the forked junction and unwind it (*Figure 2A* and schematics and controls in *Figure 2—figure supplement 1*; oligo sequences in *Table 1*). To determine whether the flush duplex is unwound, the oligo that forms the duplex was radiolabeled at its 5′ terminus. As shown in *Figure 2B*, CMG unwinds the 5′ tailed duplex but not the (untailed) flush duplex, indicating that CMG can track on dsDNA and that the central channel is large enough to accommodate the duplex. To confirm that the flush duplex oligo was not simply reannealing after being unwound, we repeated the experiment in the presence of an unlabeled trap (*Figure 2—figure supplement 2*) and obtained similar results to those in *Figure 2B*. Hence, the central channel of CMG can accommodate dsDNA, and translocation of CMG over the internal duplex is similar to the earlier DnaB studies in which the duplex is not unwound (*Kaplan, 2000*). This result is also consistent with cellular studies demonstrating that CMG forms at origins without detection of ssDNA, implying CMG encircles dsDNA at the origin (*van Deursen et al., 2012*; *Watase et al., 2012*).

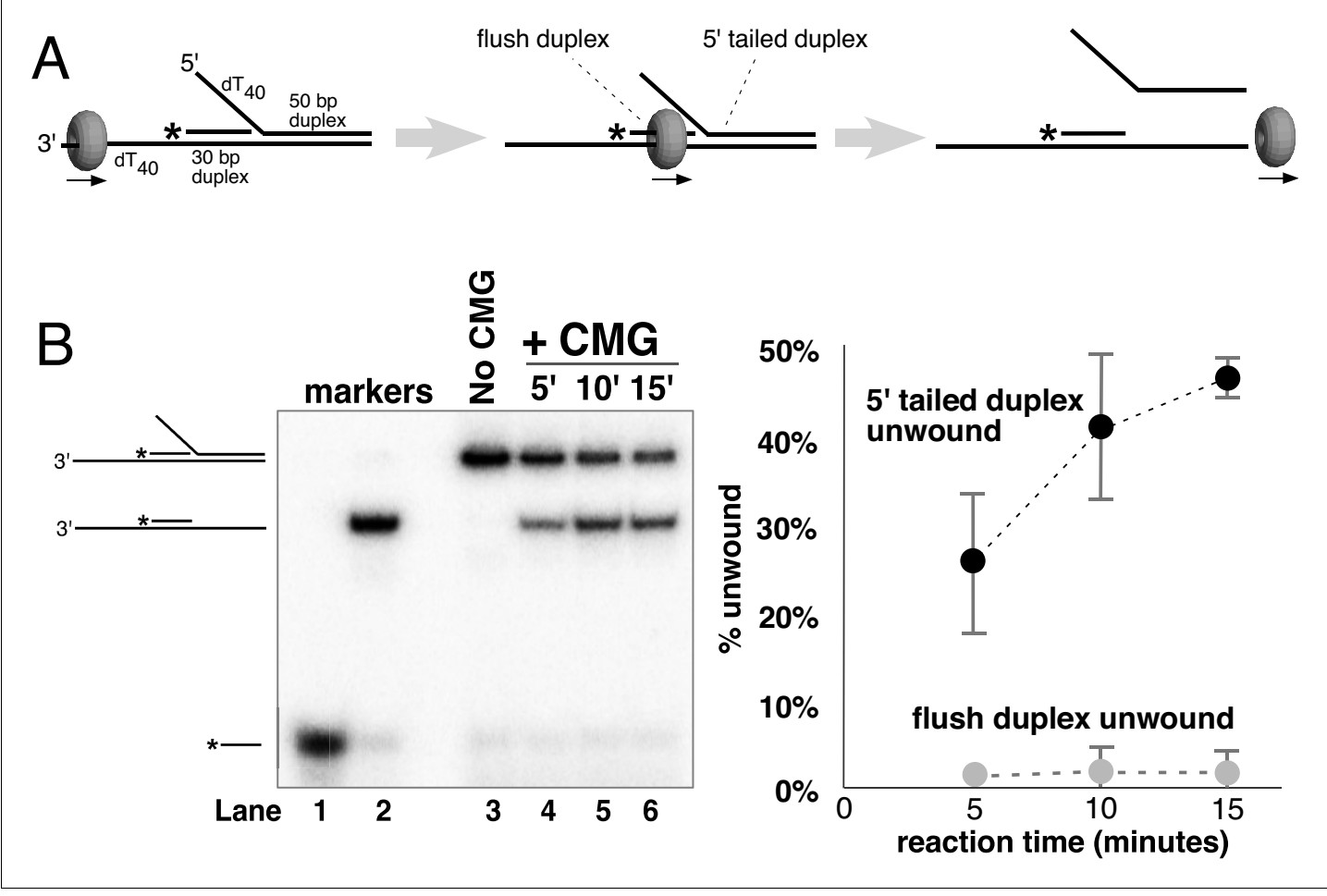

**Figure 2.** CMG translocates over flush duplex DNA without unwinding. (A) The substrate contains a 3' $dT_{40}$ ssDNA tail for CMG loading and a 5'-$^{32}$P labeled (*) flush duplex adjacent to a 5' $dT_{40}$ tailed duplex. CMG (grey ring) tracks 3'−5' along ssDNA as indicated by the arrow in the schematic, unwinding the tailed duplex (right) but leaving the flush duplex in place. A detailed description of the substrate is shown in *Figure 2—figure supplement 1*; oligo sequences are in *Table 1*. (B) Left: native PAGE analysis of the CMG unwinding reaction. See Materials and methods for reaction conditions and details. Markers (lanes 1–3) show the positions of the species indicated to the left. Right: The plot shows the time course of unwinding of the tailed duplex (dark circles) and flush duplex (light circles). Values are the average of three independent experiments and the error bars show the standard deviation. Also see *Figure 2—figure supplement 2*.

The following figure supplements are available for figure 2:

**Figure supplement 1.** CMG requires a 3' $dT_{40}$/ssDNA tail for loading.

**Figure supplement 2.** Repeat of Experiment from *Figure 2B* in the presence of a trap to prevent reannealing of flush duplex oligo.

## Dual streptavidin blocks on either strand stop CMG unwinding

To further investigate the mechanism of DNA unwinding by CMG, we used a similar substrate with a single 50-mer duplex region and $dT_{40}$ 5' and 3' tails at one end of the duplex (oligo sequences in *Table 1*; schematics at the top of *Figure 3* and in *Figure 3—figure supplement 1*). In this configuration, the 5' tail is equivalent to the lagging strand template at a replication fork and the 3' tail is equivalent to the leading strand template. A time course of CMG activity on this substrate showed a linear rate of unwinding over the first 20 min (*Figure 3—figure supplement 2*). We also determined the rate at which the separated single strands re-annealed upon mixing at 30°C. Under the conditions of the assays performed here no detectable spontaneous annealing was observed over a 20-min time course, and therefore, no trap oligonucleotide was needed to quench helicase reactions

**Table 1.** Oligonucleotides used in this study. All oligonucleotides used in this study were ordered from IDT with the indicated modifications.

| Oligo name | Sequence (5' to 3') | Modification(s) |
|---|---|---|
| Paired duplex LEAD + 3' tail | GAGACCGAACGATCCTGTAATGTCCTAG CAAGCCAGAATTCGGCAGCGTCGCGATC TGCAGCCTTGCCAGAAATCTAGTGTTTT TTTTTTTTTTTTTTTTTTTTTTTTTTTTTT | - |
| Paired duplex LEAD no 3' tail | GAGACCGAACGATCCTGTAATGTCCTAG CAAGCCAGAATTCGGCAGCGTCGCGATC TGCAGCCTTGCCAGAAATCTAGTG | - |
| flush duplex LAG | CACTAGATTTCTGGCAAGGCTGCAGATCGC | - |
| 50duplex LAG | TTTTTTTTTTTTTTTTTTTTTTTTTTTTTTT TTTTTTTTTGACGCTGCCGAATTCTGGCTT GCTAGGACATTACAGGATCGTTCGGTCTC | - |
| 50duplex LAG single biotin | TTTTTTTTTTTTTTTTTTTTTTTTTTTTTTT TTTTTTTTTGACGCTGCCGAA**T**TCTGGCTT GCTAGGACATTACAGGATCGTTCGGTCTC | Single biotin-modified dT nucleotide in **BOLD** |
| 50duplex LAG dual biotin | TTTTTTTTTTTTTTTTTTTTTTTTTTTTTTT TTTTTTTTTGACGCTGCCGAA**T**TCTGGC**T**TG CTAGGACATTACAGGATCGTTCGGTCTC | Two biotin-modified dT nucleotides in **BOLD** |
| 50duplex LEAD | GAGACCGAACGATCCTGTAATGTCCTAG CAAGCCAGAATTCGGCAGCGTCTTTTTT TTTTTTTTTTTTTTTTTTTTTTTTTT*T*T*T*T*T*T | The six dT nucleotides at the 3' end are connected by phosphorothioate bonds (*) |
| 50duplex LEAD single biotin | GAGACCGAACGATCCTGTAATGTCCTAG CAAGCCAGAA**T**TCGGCAGCGTCTTTTTT TTTTTTTTTTTTTTTTTTTTTTTTTT*T*T*T*T*T*T | Single biotin-modified dT in **BOLD**; the six dT nucelotides at the 3' end are connected by phosphorothioate bonds (*) |
| 50duplex LEAD dual biotin | GAGACCGAACGATCCTGTAATGTCC**T**AG CAAGCCAGAA**T**TCGGCAGCGTCTTTTTT TTTTTTTTTTTTTTTTTTTTTTTTTT*T*T*T*T*T*T | Two biotin-modified dT's in **BOLD**; the six dT nucleotides at the 3' end are connected by phosphorothioate bonds (*) |
| M.HpaII LAG | TTTTTTTTTTTTTTTTTTTTTTTTTTTTTTTTTTTTTTTTTGACGCTGC-(5-F-dC)-GGATTCTGGCTTGCTAGGACATTACAGGATCGTTCGGTCTC | The position of a 5-fluorodeoxycytidine is indicated by (5-F-dC) |
| M.HpaII LEAD | GAGACCGAACGATCCTGTAATGTCCTAG CAAGCCAGAATCCGGCAGCGTCTTTTTT TTTTTTTTTTTTTTTTTTTTTTTTTT*T*T*T*T*T*T | The six dT nucleotides at the 3' end are connected by phosphorothioate bonds (*) |

(*Figure 3—figure supplement 3*). Based on the results of these experiments, we chose to examine helicase activity over a 10' time course using an amount of CMG (20 nM) that gives approximately 30% unwinding after 10' to assure that the rate of unwinding is linear with respect to time. To study the effect of strand-specific blocks on CMG unwinding, we used substrates in which the small molecule biotin is covalently linked to two internal dTTP nucleotides on the duplex portion of either the leading or lagging strand template (*Figure 3—figure supplement 1*). Biotin-modified oligonucleotides are extremely tightly bound by a 53 kDa recombinant form of the streptavidin protein originally isolated from *Streptomyces avidinii* (*Green, 1990*). Control experiments showed that CMG was equally active on biotinylated substrates as on unmodified substrates in the absence of streptavidin (*Figure 3—figure supplement 4*). Furthermore, streptavidin had no effect on CMG unwinding of non-biotinylated substrates (*Figure 3—figure supplement 5*).

CMG tracks 3' to 5' on ssDNA (*Kang et al., 2012*; *Moyer et al., 2006*) and therefore addition of streptavidin to a fork substrate with dual biotin-dT in the tracking strand is predicted to block CMG because it must track along the leading strand template regardless of whether unwinding takes place outside (steric exclusion) or inside the helicase (i.e. as the CMG-forked DNA structure reveals), and indeed this was the case (*Figure 3A*, left). Based on the average of three separate experiments, CMG unwound 30% of the substrate after 10' in the absence of streptavidin and about 2% of the substrate was unwound in the presence of streptavidin (*Figure 3A*, left). These results are expected for a 3'−5' ssDNA translocase that tracks along the leading strand and they confirm that the dual biotin-streptavidin block is a strong impediment to CMG unwinding, as previously observed in *Xenopus* egg extracts (*Fu et al., 2011*).

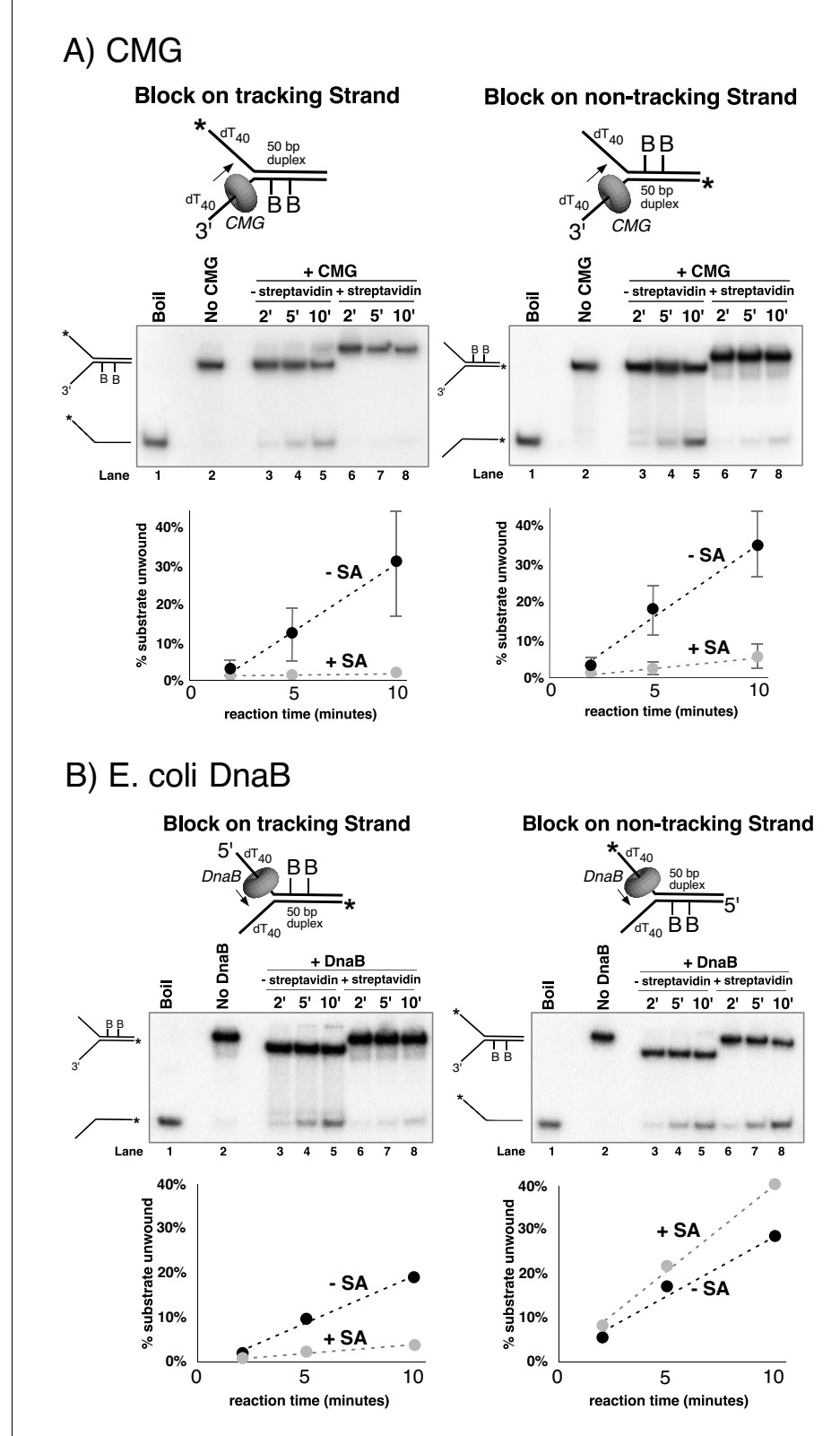

**Figure 3.** Dual biotin-streptavidin on either strand is a strong block to CMG unwinding. (**A**) Effect of dual biotin blocks on CMG. CMG was mixed with dual biotinylated DNA fork and ATP in the absence (lanes 3–5) or presence (lanes 6–8) of streptavidin. The reaction is described in Materials and methods, and the substrates are shown in *Figure 3—figure supplement 1* and in schematic above the gels. CMG (ring in the schematic) tracks 3′−5′ as
*Figure 3 continued on next page*

*Figure 3 continued*

indicated by the arrow. The radiolabeled strand is indicated by a * at its 5' end. Lane 1 shows the position of the unwound radiolabeled strand (by boiling) and lane 2 shows the forked DNA. The plots below the gels show % substrate unwound at the 2', 5' and 10' time points in the absence (dark circles) and presence (light circles) of streptavidin. Values are the average of three independent experiments and the error bars show the standard deviation. The dotted line is a linear least squares fit of the data. *Left*: the leading strand template contains dual biotin. *Right*: the lagging strand template contains dual biotin. Also see *Figure 3—figure supplements 1–6*. (B) Effect of dual biotin blocks on *E. coli* DnaB. As a control, we used *E. coli* DnaB, known to act by classic steric exclusion/external unwinding (*Kaplan, 2000*; *Kaplan et al., 2003*). DnaB translocates 5'−3' on ssDNA, placing it on the lagging strand. Left: dual biotin on the lagging strand is a strong block to DnaB unwinding when streptavidin is present (lanes 6–8) compared to the no streptavidin control (lanes 3–5). Right: dual biotin on the leading strand does not inhibit DnaB in the presence (lanes 6–8) or absence of streptavidin (lanes 3–5).

The following figure supplements are available for figure 3:

**Figure supplement 1.** Schematics of biotinylated DNA fork substrates.

**Figure supplement 2.** Time course of CMG unwinding on forked duplex.

**Figure supplement 3.** Substrate single strands do not spontaneously reanneal at 30℃.

**Figure supplement 4.** Biotinylation of the substrate does not affect CMG unwinding in the absence of streptavidin.

**Figure supplement 5.** Streptavidin does not affect CMG unwinding of a non-biotinylated substrate.

**Figure supplement 6.** Monovalent streptavidin (SA) tetramer inhibits CMG unwinding to the same extent as tetravalent streptavidin on the substrate with dual biotin on the lagging strand.

A lagging strand template block provides the test that distinguishes between external unwinding (steric exclusion) and internal unwinding because a lagging strand block should pass outside of CMG and not impede helicase activity if CMG acts by steric exclusion. If, on the other hand, the dsDNA must enter the central channel of CMG before it is unwound as indicated by the CMG-forked DNA structure, then a lagging strand block should halt the progress of CMG about 10 bp upstream of the biotinylated nucleotide (*Fu et al., 2011*). Accordingly, we repeated the experiments with dual biotin-dT on the lagging strand template portion of the duplex. Surprisingly, addition of streptavidin reduced CMG unwinding to almost the same extent as with a leading strand template block (*Figure 3A*, right), with only 5% of the substrate unwound after 10' in the presence of streptavidin compared to 35% in the absence of streptavidin. This result is most compatible with the structure of CMG-forked DNA where both strands enter the central channel of the CMG ring (*Georgescu et al., 2017*). Assuming this interpretation is correct, it suggests that dsDNA entry into the N-tier of CMG is important to the unwinding mechanism of CMG. However, there are two alternative explanations as to why a lagging strand block might slow down a helicase. First, the lagging strand might track in a groove on the outside of CMG as suggested by cross-linking studies (*Graham et al., 2011*; *Petojevic et al., 2015*) and the tight binding of streptavidin to biotin on DNA may block this track. Second, the lagging strand might pass through a separate ring in CMG formed by the accessory proteins GINS/Cdc45 at the side of the Mcm2-7 ring, as first observed in *Drosophila* CMG. It remains to be seen whether this ring is large enough to accommodate ssDNA in other species, but the high-resolution structure of *S. cerevisiae* CMG indicates that the opening formed by the accessory factors that was observed in lower resolution structures was filled in yeast CMG (*Yuan et al., 2016*).

The two lagging strand biotins were seven bases apart and thus may act independently and bind a separate streptavidin. But to address the possibility that a single streptavidin tetramer could bind both biotins and distort the duplex, thereby somehow inhibiting a steric exclusion process, we performed the experiment using a re-engineered mutant form of streptavidin in which only one subunit of the tetramer can bind to biotin (*Howarth et al., 2006*). The results show that addition of this

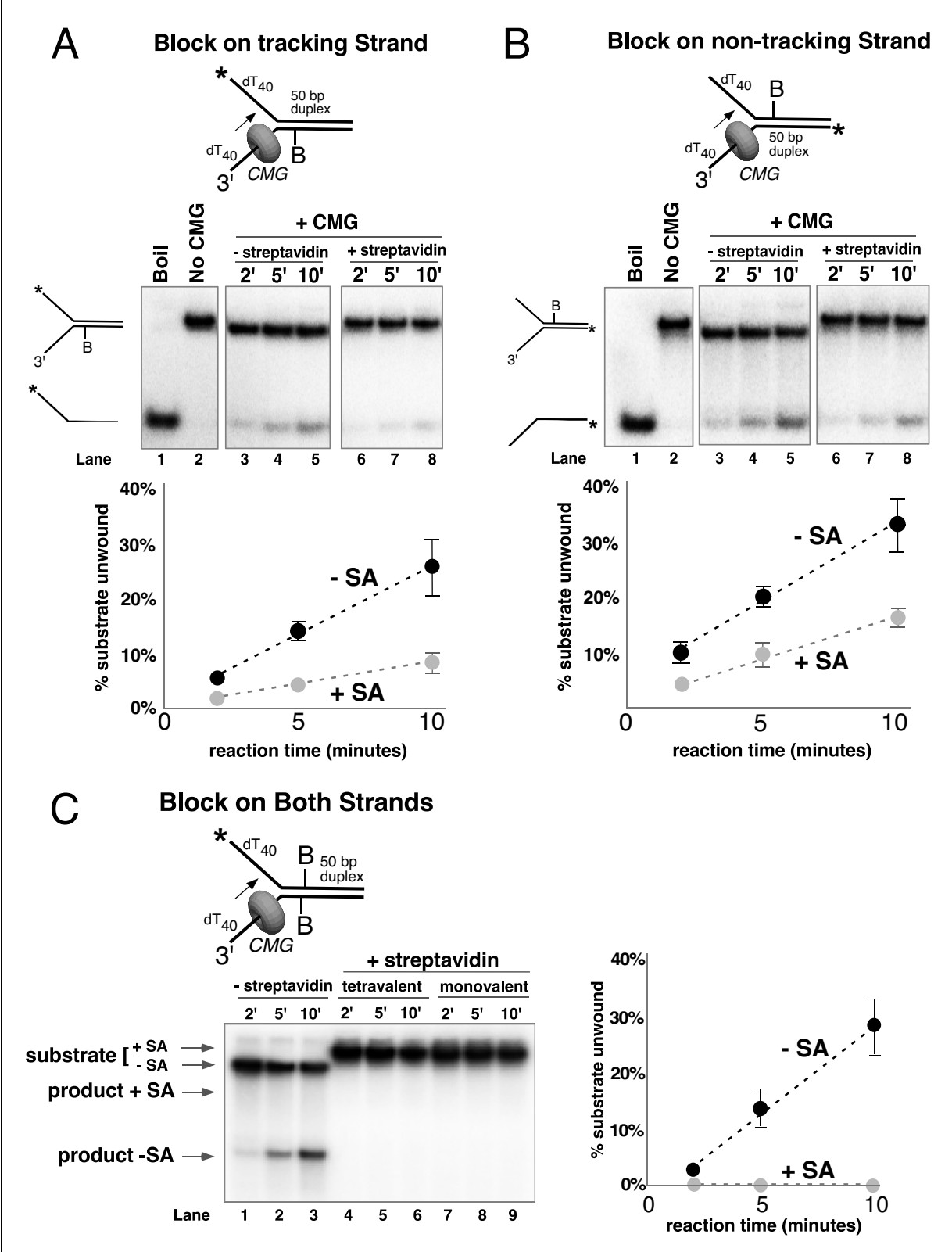

**Figure 4.** Single biotin-streptavidin is a weaker block to CMG unwinding on either strand but not when placed on both strands. Reaction conditions and analysis are identical to those in *Figure 3* except that the substrate contained a single biotin on the leading strand (**A**), lagging strand (**B**), or both (**C**) as indicated in the schematics above the gels; and the substrate in (**A**) and (**B**) was pre-incubated with 2 μg/ml streptavidin before addition of CMG instead of 4 μg/ml. The substrates are described in detail in *Figure 3—figure supplement 1*.

'monovalent' streptavidin blocked CMG unwinding to the same extent as wild-type (tetravalent) streptavidin (*Figure 3—figure supplement 6*), indicating that distortion of the duplex is not responsible for the observed block to unwinding. As an additional control, we examined the effect of a dual biotin block on unwinding by the homohexameric DnaB helicase of *E. coli* (*Figure 3B*). DnaB tracks with the opposite polarity to CMG, 5′−3′ on ssDNA, placing it on the lagging strand template. Unwinding by DnaB was previously shown to be blocked by a single biotin-streptavidin on the tracking strand (lagging strand template) but was not inhibited at all by streptavidin on the non-tracking strand (leading strand template) (*Kaplan, 2000*; *Kaplan et al., 2003*). As shown in *Figure 3B,a* dual biotin-streptavidin on the tracking strand (lagging strand template, left) strongly inhibited DnaB, whereas a dual biotin-streptavidin block on the non-tracking strand (leading strand template, right) did not inhibit unwinding by DnaB. This result indicates that a dual biotin/streptavidin block is not an impediment to a hexameric helicase that operates by steric exclusion and supports the conclusion that CMG does not act by a classic steric exclusion mechanism, as indicated by the CMG-forked DNA structure.

## CMG can dislodge a single biotin-streptavidin block from the leading strand

To further understand CMG behavior during unwinding, we examined the effect of a weaker single biotin-streptavidin block on unwinding by CMG instead of the stronger dual biotin-streptavidin block. As noted above, other hexameric helicases like DnaB are completely blocked by a single streptavidin binding to biotin on the tracking strand (*Kaplan, 2000*; *Kaplan et al., 2003*). CMG was, in fact, inhibited by a single biotin block, but to our surprise, it was not a very stringent block (*Figure 4*). With a single biotin on the leading (tracking) strand, unwinding at 10′ was reduced from a normal level of 25% in the absence of streptavidin to a level of 8% in the presence of streptavidin (*Figure 4A*); a single biotin-streptavidin block on the lagging strand template reduced activity from 33% to 16% (*Figure 4B*). The greater inhibition by a dual block observed in *Figure 3A* is essentially the additive behavior of two independent blocks. In other words, for the lagging strand, unwinding is reduced by approximately half by the presence of the single block (from 33% to 16% at 10 min). If a second block also reduces unwinding by half, this would bring it from 16% to 8%, which is close to the observed level of 5% (*Figure 3A*). Similarly, on the leading strand, a single block reduces unwinding by two-thirds, from 25% to 8% unwound at 10 min. A further two-thirds reduction by a second block would bring the level of unwinding to ~2.5% which is the level observed in *Figure 3A* for the dual leading strand block. We sought to determine whether this additive effect would apply to single blocks placed on both strands in adjacent positions, and indeed, the combination of single biotin blocks on both strands was strongly inhibitory to CMG unwinding in the presence of streptavidin (*Figure 4C*, lanes 4–6). To assure that inhibition was not attributable to cross-linkage of the two strands by a single tetramer of streptavidin, we repeated the experiment using monovalent streptavidin and observed the same strong inhibition of CMG unwinding (*Figure 4C* lanes 7–9).

The greater ability of CMG to proceed past single biotin-streptavidin blocks on either strand, compared to dual biotin blocks, provided the opportunity to analyze the unwound products and to determine if streptavidin was displaced from biotin or bypassed and left on DNA. To do so, we modified the assay in two ways. First, we radiolabeled the strand containing the biotin and determined that streptavidin causes a gel shift at a clearly distinguishable position from unbound DNA in a native PAGE gel (*Figure 5* and *Figure 5—figure supplement 1*). Second, we pre-incubated the substrate with streptavidin and then added a 20-fold molar excess of free biotin along with CMG on ice before starting the reaction at 30°C. Under these conditions, if streptavidin is displaced, it will bind the excess free biotin and the radiolabeled DNA product should no longer shift in the gel (i.e. it will migrate in the gel as ssDNA without bound streptavidin). Control experiments showed that the free biotin trap completely prevents binding of streptavidin to the biotinylated oligo when added before streptavidin but not when added after streptavidin (*Figure 5—figure supplement 1*). These results also establish that spontaneous dissociation of streptavidin is negligible during the timeframe of the assay (*Figure 5—figure supplement 1*).

CMG tracks along the leading strand, which has been shown to pass through the central channel of CMG. Thus, we expected that the only way CMG could bypass a biotin-streptavidin block on the leading strand was by removing it. The result, using excess biotin, showed the expected ~70% inhibition of unwinding as observed in *Figure 4A*, and examination of the unwound products revealed

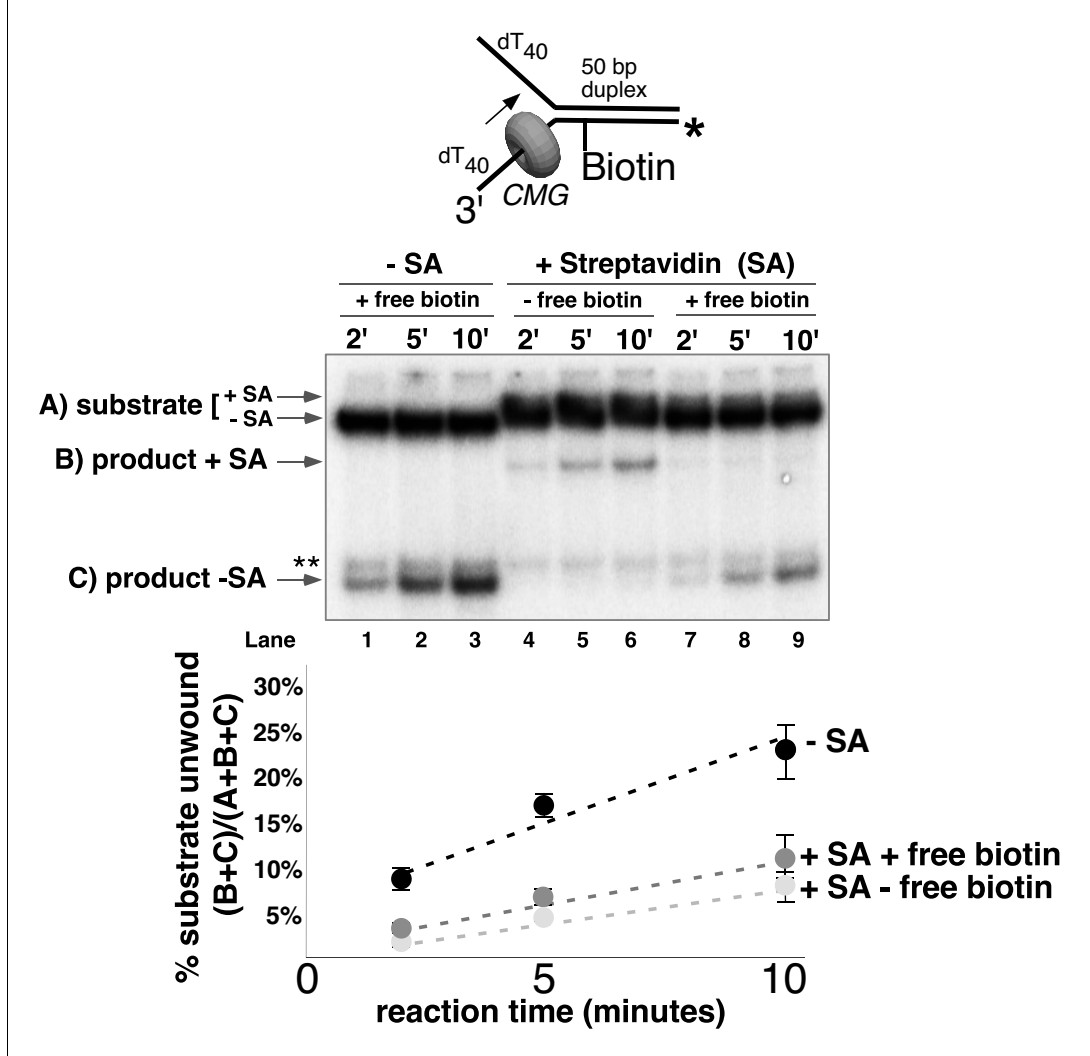

**Figure 5.** CMG can displace streptavidin from biotin on the leading strand. In the reaction shown in lanes 4–9, the substrate was pre-incubated with 2 µg/ml streptavidin for 5' at 30°C and then placed on ice for 10' before addition of CMG only (lanes 4–6) or CMG plus D-biotin to a final concentration of 750 nM (lanes 7–9) as a trap for streptavidin displaced from biotin by CMG. The reaction in lanes 1–3 was performed in the absence of streptavidin. As shown in the schematic above the gel, in these experiments the radiolabel (*) was on the leading strand template containing the biotin-dTTP and when unwound it migrates at different positions in the gel in the presence (**B**) or absence (**C**) of streptavidin (SA) as indicated to the left of the gel. The band at the position indicated by ** to the left of the gel is a background band that is observed when the biotinylated strand is radiolabeled. Below the gel is a plot of the time course of unwinding for: lanes 1–3 (-SA), lanes 4–6 (+SA – free biotin), and lanes 7–9 (+SA + free biotin) showing the percent unwound product relative to total DNA (product plus unreacted substrate). Values are the average of three independent experiments and the error bars show the standard deviation. Displacement of streptavidin by CMG is revealed by a down shift in the migration of the unwound product from the product +SA position (**B**) in the absence of the D-biotin trap (lanes 4–6) to the product –SA position (**C**) in the presence of the trap (lanes 7–9). Also see *Figure 5—figure supplement 1*.

The following figure supplement is available for figure 5:

**Figure supplement 1.** Free biotin trap prevents binding of streptavidin to biotinylated oligo when added before streptavidin but not when added after streptavidin.

that streptavidin was indeed displaced from all the strands that were unwound (*Figure 5*, compare the migration of the unwound product in lanes 7–9 with that in lanes 4–6). Hence, CMG translocates with sufficient force to displace streptavidin from DNA while unwinding, as shown previously for the hexameric SV40 T-antigen helicase (*Morris et al., 2002*). This is in stark contrast to *E. coli* DnaB which lacks the ability to get past a single biotin-streptavidin block (*Kaplan, 2000*; *Kaplan et al.,*

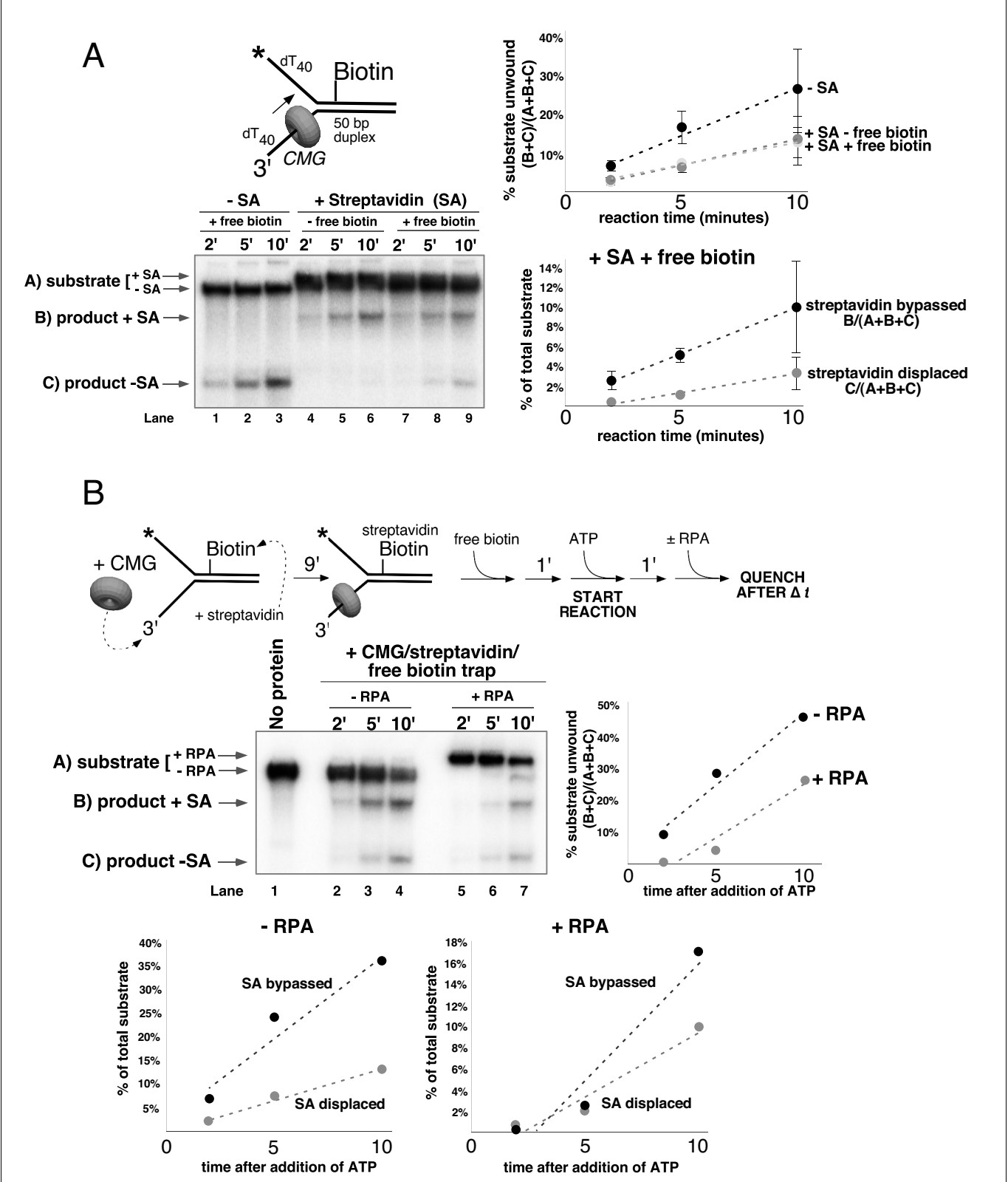

**Figure 6.** CMG can bypass or displace streptavidin from the lagging strand. (A) Reaction conditions and analysis are identical to those in *Figure 5* except the substrate was radiolabeled (*) on the biotinylated lagging strand template as shown in the schematic above the gel. Reactions in the absence of biotin trap (lanes 4–6) show the upshifted unwound product DNA. Reactions in the presence of biotin trap (lanes 7–9) show two products. One product is in the upshifted position and thus still contains streptavidin, and the other product is downshifted to the position of unwound substrate

*Figure 6 continued on next page*

*Figure 6 continued*

lacking bound streptavidin (SA displaced). The plot at the top right shows total unwinding in the absence of streptavidin (lanes 1–3) and in the presence of streptavidin with free biotin (lanes 7–9) or without (lanes 4–6). Values are the average of three independent experiments and the error bars show the standard deviation. The plot at the bottom right shows the time course of appearance of the distinct SA bypassed and SA displaced products in lanes 7–9. Also see *Figure 6—figure supplements 1–2*. (B) Single hit experiment using RPA to prevent reinitiation of CMG loading during the assay. Top: scheme of the experiment. CMG is preincubated 9 min with the forked DNA having a single lagging strand biotin-streptavidin block, then biotin trap is added, followed by ATP, and RPA is added 1 min later. The gel, below, shows equivalent reactions except RPA was not added to the reaction in lanes 2–4. Comparison of reactions without RPA (lanes 2–4) with reactions containing RPA (lanes 5–7) shows that RPA blocks reinitiation, as demonstrated previously (*Georgescu et al., 2014*). The plots below the gel show the quantitation of the two products formed in reactions minus RPA (left) and plus RPA (right).

The following figure supplements are available for figure 6:

**Figure supplement 1.** CMG displacement of streptavidin from ssDNA.

**Figure supplement 2.** Helicase reactions by CMG on a single lagging strand biotinylated fork using two concentrations of CMG.

*2003*). In overview, the results with leading strand blocks indicate that CMG is inhibited by the leading strand streptavidin block in most cases, but when CMG succeeds in unwinding past the block it does so by displacing the leading strand streptavidin from biotin-DNA. As described earlier, when two streptavidin blocks are closely positioned on the leading strand, the likelihood of displacing them both is additive which accounts for the strong inhibition of CMG unwinding by a dual biotin-streptavidin block (*Figure 3A*) (*Fu et al., 2011*).

## CMG can take either of two paths to bypass a lagging strand biotin-streptavidin block

Next, we performed similar experiments using a single streptavidin block on the duplex portion of the radiolabeled lagging strand (*Figure 6*). Again, we see approximately 50% inhibition of unwinding as observed in *Figure 4B*, and when excess biotin is not present in the reaction the unwound lagging strands migrate at the shifted position in the gel as expected (*Figure 6A*, lanes 4–6). However, when excess biotin is added to the reaction as a trap for displaced streptavidin, we observe two products, with ~3/4 of the unwound strands migrating at the shifted position (with streptavidin still bound) and the remaining ~¼ migrating at the non-shifted position indicating displacement of the bound streptavidin (*Figure 6A*, lanes 7–9).

Formation of two products can be explained by one mechanism, which we propose first, although we also entertain alternative explanations below. Insight into a single mechanism that may explain the two lagging strand products is derived from the cryoEM structure of CMG bound to forked DNA (*Figure 1A*) (*Georgescu et al., 2017*). Both strands of DNA enter the central channel of CMG, and therefore, CMG is sterically obstructed by the lagging strand streptavidin block a short distance before reaching the biotinylated nucleotide (as observed in the structure). This situation likely causes CMG to pause unwinding while it attempts to translocate past the streptavidin, explaining the approximately 50% inhibition of CMG in *Figure 4B* and *Figure 6A*. When CMG succeeds in unwinding the DNA, pushing against the streptavidin block can sometimes (i.e. about 25% of the time) lead to displacement of the lagging strand streptavidin in the same way the leading strand streptavidin is displaced, but the bulk of the time (about 75%) CMG bypasses the block and the streptavidin is retained on DNA. One can entertain several models, consistent with the structure, in which CMG might bypass the streptavidin block without displacing it, including models that retain an internal DNA unwinding point, and these are presented in the Discussion. Importantly, regardless of the actual mechanism of bypass, the results largely follow a classic steric exclusion process in which the streptavidin is retained on the unwound DNA.

Although streptavidin was removed from only ~¼ of the unwound lagging strands, it is important to note that removal of a block on the non-tracking strand is a key finding in support of an internal unwinding mode for CMG that is without clear precedent in studies of other hexameric helicases so we performed extensive controls to rule out alternative explanations for this result. For example, another explanation for removal of streptavidin from the lagging strand is that CMG first unwinds

the duplex DNA without streptavidin displacement (i.e. steric exclusion) and then reloads onto the unwound lagging strand ssDNA, which would then become the tracking strand from which CMG could remove streptavidin. To assess this possibility, we examined the ability of CMG to remove streptavidin from unannealed lagging strand template ssDNA (*Figure 6—figure supplement 1A*). The time course shows that CMG does not detectably remove streptavidin after 10 min, the time-frame of the assays in *Figure 6*. We presume this is because CMG loads poorly onto the 3' end of the lagging strand oligo because it does not have a 3' poly-dT helicase loading sequence that was previously shown to be essential for CMG self-loading (*Kang et al., 2012*) (confirmed in *Figure 2—figure supplement 1*).

To further demonstrate that both unwound products (streptavidin retained and streptavidin displaced) arise in the process of single CMG unwinding events and not as a series of separate events, we performed a 'single-hit' assay in which we preincubated CMG with the DNA and then used RPA as a blocking agent to prevent additional CMG reloading onto DNA during the assay, as illustrated in *Figure 6B*. We previously showed that RPA prevents CMG from loading onto a fork substrate by competitively binding the ssDNA tails but does not displace CMG that has already loaded onto DNA in a preincubation step (*Georgescu et al., 2014*). Accordingly, we pre-incubated CMG with the DNA substrate in the presence of streptavidin (without RPA or ATP) for 10'. One minute before adding ATP, excess free biotin was added as a trap for displaced streptavidin and one minute after starting the reaction, RPA was added to block further CMG loading (see reaction scheme at the top of *Figure 6B*). We cannot exclude the possibility that some additional CMG loads on the substrate in this 1-min interval, but our previous studies show that RPA limits further CMG loading. We used a higher concentration of CMG (45 nM) in this experiment to assure that unwinding in the presence of RPA would be sufficient to determine streptavidin retention and displacement, and accordingly total unwinding was higher than in *Figure 6A*. Under these conditions, even when RPA is present to prevent CMG reloading onto the unwound ssDNA, streptavidin displacement was still observed at levels comparable to those seen in *Figure 6A*. Reactions in the absence of RPA give more unwinding overall in this experiment, as one would expect for a reaction that does not limit CMG loading during the assay. The result of this single-hit assay, where RPA is present to prevent reloading of CMG onto unwound DNA, supports the conclusion that both of the DNA unwound products (i.e. streptavidin bound and streptavidin displaced) are the result of single CMG unwinding events, in which CMG either bypassed or removed streptavidin from the lagging strand to unwind the duplex.

To further support the conclusion that streptavidin is being removed by CMG during unwinding and not by some non-specific mode of binding (i.e. binding to the external surface of CMG), we repeated the lagging strand biotin-streptavidin experiment with two different concentrations of CMG (15 nM and 60 nM) in the presence of the free biotin trap (*Figure 6—figure supplement 2*). Increasing the CMG concentration fourfold also increased unwinding by approximately fourfold as expected (graph at left in *Figure 6—figure supplement 2*). At both concentrations, the two products increase with time (graph at right in *Figure 6—figure supplement 2*) but the ratio of streptavidin bound to streptavidin displaced remains the same. If CMG was removing streptavidin by some non-specific association with the substrate and not during unwinding, the proportion of unwound strands with displaced streptavidin should have been much higher with the higher concentration of CMG but it was similar at the two concentrations of CMG.

## A covalent lagging strand protein-DNA adduct forms a stringent block to CMG

The experiments thus far demonstrate that CMG can displace streptavidin from DNA; however, streptavidin is not specifically bound to the DNA but instead is bound to a biotin that is attached to DNA by a chemical linker. Therefore, we wished to form a covalent protein adduct on the lagging strand to analyze CMG bypass of a block that cannot be displaced. The bacterial *Hpa*II methyltransferase (*M.Hpa*II) has been shown to form a covalent adduct to DNA when 5-fluorodeoxycytidine (5-FDC) is present in the second position of its 4 bp recognition sequence (*Chen et al., 1991*; *Duxin et al., 2014*). Hence, we placed the 5-FDC/*Hpa*II site in the duplex stem on the lagging strand template and formed a covalent *M.Hpa*II-forked DNA adduct to assess whether CMG could bypass the adduct (see schematic in *Figure 7A*). The crystal structure of the closely related *Hha*I methyltransferase (83% identity) shows that when the enzyme is covalently attached to it target base in this fashion it also remains bound to its recognition sequence on the DNA and thus would be expected

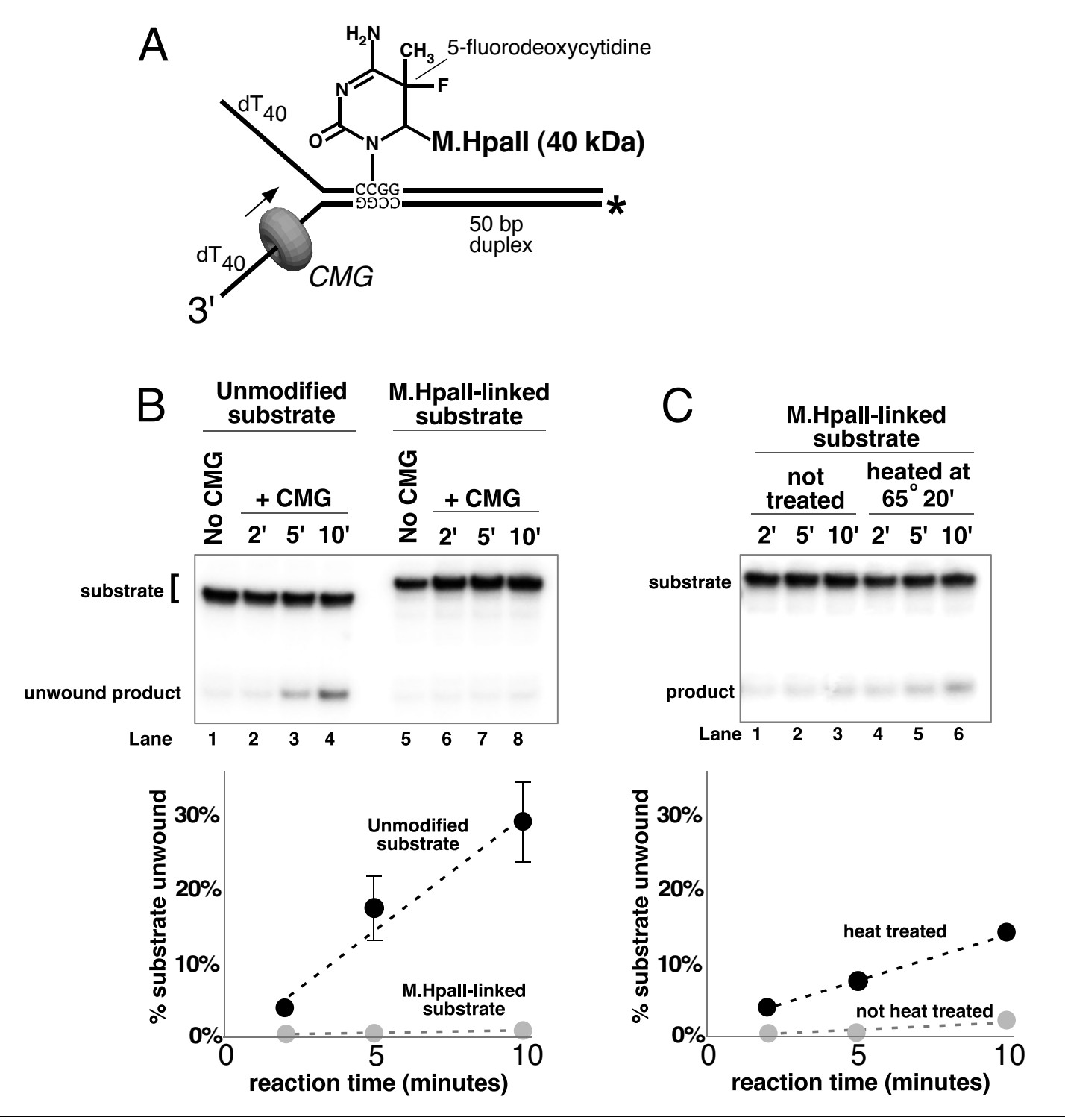

**Figure 7.** A covalent lagging strand protein-DNA adduct forms a stringent block to CMG. (**A**) Schematic of the substrate used in these reactions. The duplex portion of the fork contains the 4-base recognition sequence for the *Hpa*II methyltransferase. Replacement of the second dCTP in the recognition site with 5-fluorodeoxycytidine (5-FDC) traps a covalent intermediate in the methylation reaction in which *M.Hpa*II remains bound to the nucleotide base on the lagging strand as shown (***Chen et al., 1991***). Sequences of the oligos are in *Table 1*. (**B**) CMG unwinding reactions using the 5-FDC substrate with (lanes 6–8) or without (lanes 2–4) *M.Hpa*II modification. Except for the substrate, reactions conditions are the same as those of *Figure 3* (no streptavidin). (**C**) The *M.Hpa*II modified 5-FDC substrate was heated at 65° for 20' to inactivate *M.Hpa*II, cooled on ice, and added to a CMG unwinding reaction (lanes 4–6) identical to that in (**B**) except 40 nM CMG was used. Lanes 1–3 show unwinding of the untreated substrate under the same conditions.

to provide a highly stringent block for proteins that translocate on DNA (*Klimasauskas et al., 1994*). Despite this irreversible and close attachment of *M.Hpa*II to DNA, studies in the *Xenopus* egg extract system have demonstrated that replication forks can bypass this block and that leading strand *M.Hpa*II blocks are bypassed by a process in which a protease in the extract proteolytically digests *M.Hpa*II to enable fork progression (*Duxin et al., 2014*). The results using pure CMG demonstrate that isolated CMG cannot bypass the *M.Hpa*II-lagging strand adduct (*Figure 7B*). Considering that CMG can bypass a single streptavidin block on the lagging strand, the result suggests that the very close covalent attachment of a DNA-binding protein on the lagging strand cannot be bypassed by CMG. To test this, we heat treated the covalent *M.Hpa*II adduct to denature/inactivate the protein, possibly providing more flexibility of attachment and thus enabling bypass. Indeed, this treatment provided measurable bypass of the lagging strand *M.Hpa*II adduct by CMG (*Figure 7C*), although the bypass was still less efficient than in the case of a streptavidin block (*Figure 6B*).

The results of all our experiments can be explained in a simple way by the CMG-forked DNA structure (*Georgescu et al., 2017*) but would be difficult to explain by steric exclusion/external unwinding or classic side channel extrusion models. Hence, from the structure, the dsDNA enters the N-tier of CMG, and therefore one expects that blocks on either strand will result in stalling the helicase a short distance before reaching the biotinylated nucleotide or covalent adduct. In the case of streptavidin blocks, the extent of stalling reflects the amount of streptavidin that is displaced or bypassed, and thus inhibition of unwinding by a single block on the leading strand (~67% inhibition, *Figure 4A*) is greater than on the lagging strand (~50% inhibition *Figure 4B*) because there is only one route forward, streptavidin dissociation (*Figure 5*). With a block on the lagging strand, by contrast, there is less stalling of the helicase and inhibition of unwinding because CMG is able to bypass a block without removing it (*Figure 6*). The streptavidin is left on the DNA, which implies a route that proceeds faster than streptavidin displacement, as seen in the different rates at which the bypassed and removed products appear (*Figure 6A*, graph at bottom right). This result is typically associated with a steric exclusion process with an external unwinding point, but this model does not explain the significant inhibition of unwinding by a lagging strand block observed in our experiments. The results are more easily explained by a steric exclusion process with internal unwinding as seen in the CMG-forked DNA structure (*Georgescu et al., 2017*), and possible mechanisms by which a block on the lagging strand may be bypassed are presented in the Discussion.

## Discussion

The classic steric exclusion mechanism posits that the helicase encircles only one strand and tracks along it while excluding the other strand from the central channel, acting as a moving wedge to split the DNA duplex. In this model, the DNA separation point is outside of the helicase (*Figure 1B*, left). Another proposed mechanism for hexameric helicases, one that has not yet been demonstrated for any helicase, is the side channel extrusion model in which the helicase encircles both strands of DNA and the point of unwinding is internal to the central channel (*Figure 1B*, middle) In this model, as the DNA is unwound, the non-tracking strand is extruded to the outside of the helicase through a side channel that is usually depicted as an opening at subunit interfaces located at the narrow 'neck' that joins the N- and C- terminal domains of subunits in hexameric helicases (*Brewster et al., 2008*). For example, the Mcm2-7 double hexamer structure shows an opening in a location between subunits Mcm2/6 at the neck joining the N- and C-terminal domains and this was proposed to be a possible side exit channel for the lagging strand (see Extended Data *Figure 7* in [*Li et al., 2015*]).

Based on the work of this report and the structure of CMG-forked DNA (*Georgescu et al., 2017*), it seems likely that CMG operates by neither of these 'classic' mechanisms. In the absence of the CMG-forked DNA structure, the lagging strand blocking data presented here could be interpreted as a 'gated' side channel extrusion process in which CMG can either remove the block or, more frequently, bypass the block by a gate that can open to allow the blocked DNA to pass outside of the helicase. While the CMG-forked DNA structure shows that dsDNA enters the central channel and that there is an internal unwinding point of the dsDNA, consistent with the side channel extrusion model, the structure did not reveal the location of the lagging strand nor a side channel (*Figure 1A*) (*Georgescu et al., 2017*). Therefore, we do not propose a side channel extrusion process at this time, although some type of side exit channel cannot be rigorously excluded since the lagging strand is not visualized. Instead, we interpret the effect of blocks on the lagging strand in

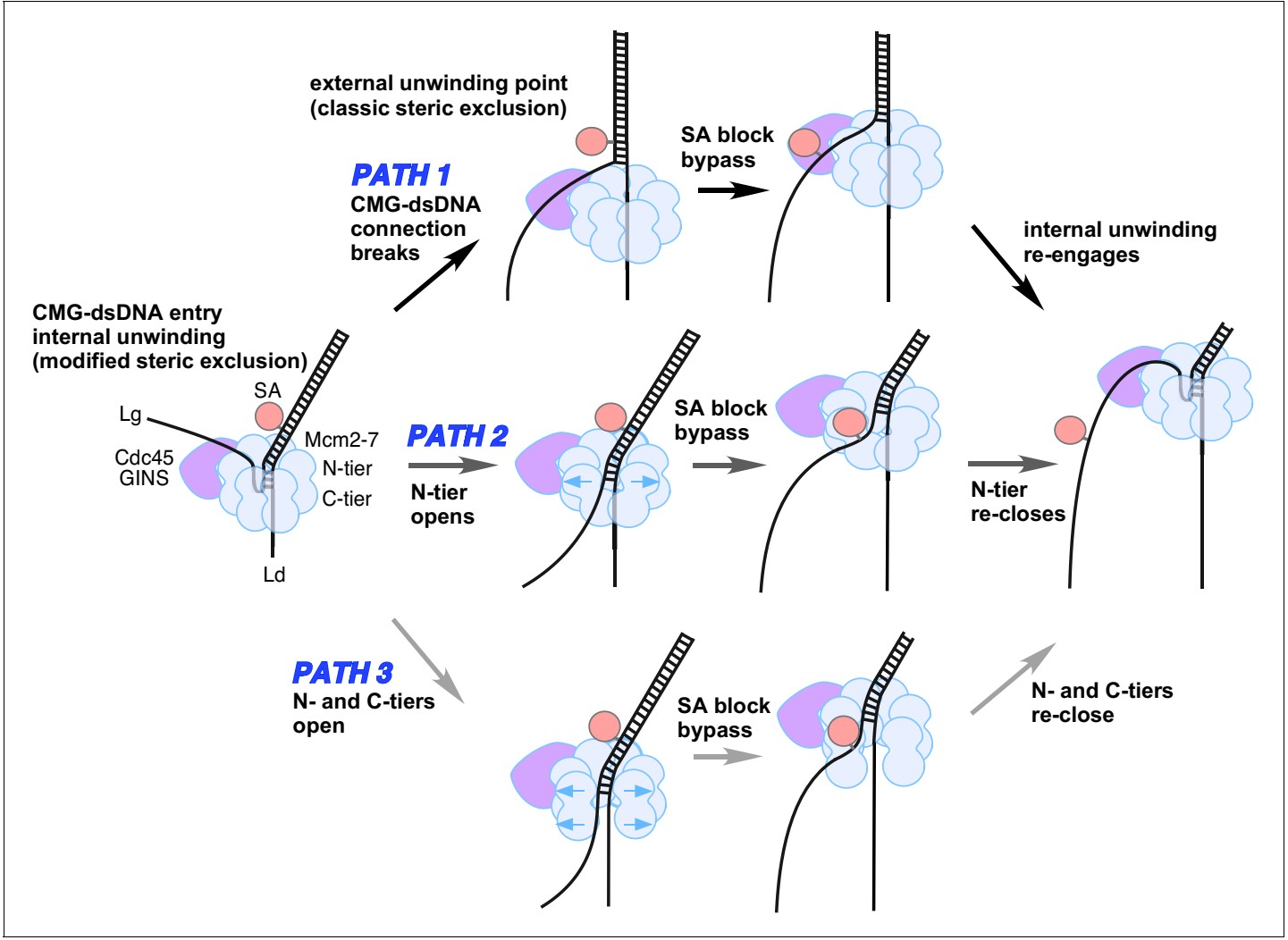

**Figure 8.** Possible paths of CMG bypassing streptavidin blocks on the lagging strand. The illustration at the far left depicts the observed structure of CMG at a fork and the proposed exit path of the lagging strand template. Path 1 illustrates conversion to a classic steric exclusion process with an external unwinding point. Paths 2 and 3 illustrate streptavidin bypass by opening of either the N-tier or both tiers of the Mcm2-7 ring. The unwinding point could remain internal, as illustrated, or could become external. The illustration at the right suggests the CMG reassumes its initial conformation after passing the block.

terms of the interaction of CMG with the dsDNA stem of the forked junction, clearly observed in the structure. The dsDNA is held at a 28° tilt relative to the axis of the central channel, which implies that the dsDNA is held tightly by CMG (*Georgescu et al., 2017*). Thus, instead of exiting by a side channel, the lagging strand might simply bend back out of the central channel and proceed through surface grooves between the zinc fingers of CMG that surround the dsDNA and this would effectively exclude the lagging strand from the central channel (*Figure 1B*, right). For this reason, we propose that CMG operates by a modified steric exclusion process in which the the dsDNA enters the central channel for unwinding but the lagging strand template is subsequently excluded to the exterior of CMG through the same opening by which it enters rather than through a side channel. This new model helps to explain how a lagging strand block can strongly inhibit CMG unwinding (*Figure 3A* and *Figure 4B*) while still allowing for bypass of the block without removing it (*Figure 6*).

## Possible mechanisms for bypass of a block on the lagging strand

We do not know the details as to how CMG bypasses a lagging strand streptavidin block, but some possibilities are illustrated in *Figure 8*. One path (Path 1, top) could be that CMG converts to a

'classic' steric exclusion helicase in which the unwinding point is outside the central channel and the dsDNA-CMG interface is broken such that dsDNA does not enter CMG. After passing the block, CMG could resume the internal unwinding mode. This path must be inefficient for isolated CMG because a single biotin-streptavidin block reduces the rate of product formation ~50% throughout the 10' time course (*Figure 4B*), indicating that CMG is slowed by the block. Nevertheless, when unwinding does proceed it most frequently occurs without removal of the streptavidin (*Figure 6*) suggesting that CMG might be able to move past the block by loosening its grip on the dsDNA, leading to external unwinding. In a smaller proportion of cases, the force exerted by CMG translocation is strong enough to disrupt the biotin-streptavidin interaction on the lagging strand which suggests that CMG retains its strong grip on the dsDNA when displacing lagging strand streptavidin.

As noted earlier, an alternative explanation for displacement of streptavidin from lagging strand biotin is that the lagging strand interacts with the outside surface of CMG, as demonstrated for *Drosophila* CMG and an archaeal MCM homohexamer by cross-linking and other studies, and that this interaction displaces the streptavidin (*Graham et al., 2011*; *Petojevic et al., 2015*). However, there is nothing in these studies that points to an interaction between CMG and ssDNA that is tight enough to disrupt the interaction between streptavidin and biotin, which is one of the strongest non-covalent interactions ever demonstrated (*Green, 1990*). Indeed, such a possibility is thus far unprecedented in any protein-DNA interaction system and, conceptually, an interaction of CMG with the excluded strand that matches or exceeds the affinity of streptavidin for biotin would most likely prevent replisome movement along DNA. How such an interaction might occur without fatally inhibiting replisome progression would require an explanation that eludes the authors of the current study.

A second possible path for bypass of a lagging strand block is shown in the middle of *Figure 8* (Path 2). In this path, the N-tier opens at a subunit interface enabling the CMG to bypass a bulky block while possibly retaining its grip on the dsDNA in the central channel. We have previously noted that the interface between the NTDs of Mcm4/6 has the least buried surface area relative to the other subunit interfaces and might be the first to open if forced by a block (*Yuan et al., 2016*).

A third path is illustrated at the bottom of *Figure 8* (Path 3). In this path, both the N-tier and C-tier open at a subunit interface, potentially enabling the CMG to bypass a bulky block on either strand. We presume this would not be the Mcm2/5 subunit interface that is initially opened for loading of Mcm2-7 onto the origin since this interface in CMG is braced by the Cdc45/GINS subunits, which have been shown to prevent leading strand escape from the Mcm2/5 gate in DNA cross-linking studies (*Petojevic et al., 2015*). However, at some point in origin activation, the CMG complexes that are formed on dsDNA and thus initially encircle dsDNA must transit to encircling ssDNA for helicase action, and this infers that a gate for ssDNA passage through both tiers of CMG in fact exists but its identity is unknown. Perhaps, this strand passage gate is used for bypassing a lagging strand block. With a complete opening of CMG, the leading strand might be more liable to becoming lost from CMG. However, it is also possible that the N-tier and C-tier do not open at the same time, thereby keeping the leading strand locked inside throughout the passage of a block. Another potential problem of this path is that there is no obvious way to keep the two strands separated, but perhaps the CMG could hold the strands apart and prevent them from reannealing.

We note that the middle and bottom paths for bypass of a lagging strand block (Paths 2 and 3), which invoke an opening in an interface, might allow the CMG to keep unwinding DNA at an internal site. But given the scarcity of details on how helicases pass blocks, one cannot distinguish between the three possible mechanisms, or whether some other mechanism is at work. Clearly, this important aspect of helicase biochemistry and structure will require further studies.

Finally, as explained in the Introduction, studies in the *Xenopus* extract system of the complete replisome show limited and often transient inhibition by lagging strand blocks (*Fu et al., 2011*), in contrast to the current study using isolated CMG. Hence, we propose that there exist other factor(s) in a complete replisome that assist CMG in bypassing lagging strand blocks and we are currently investigating candidate proteins that might serve this function. Whether the assistance comes in the form of byassing the block or displacing the block is unknown since this question was not addressed in the *Xenopus* study. However, we assume that the assistance will be in the form of bypassing lagging strand blocks without displacement since the isolated CMG takes this path in preference to displacing a block (*Figure 6*). Further studies will be needed to identify the factor(s) and understand how they modulate CMG both structurally and biochemically.

## Materials and methods

### Reagents

Radioactive nucleotides were from Perkin Elmer and unlabeled nucleotides were from GE Healthcare. DNA modification enzymes including *M.HpaII* methyltransferase were from New England Biolabs. DNA oligonucleotides were from Integrated DNA Technologies except for the oligo containing 5-fluorodeoxycytidine, which was from Bio-Synthesis (Lewisville, TX). Immunopure streptavidin was from Pierce/Thermo Scientific. Of streptavidin powder, 5 mg was resuspended in 0.5 ml distilled water to make 10 mg/ml stock. D-Biotin (50 mM aqueous solution) was from Invitrogen/Molecular Probes. Protein concentrations were determined using the Bio-Rad Bradford Protein stain and bovine serum albumin as a standard.

### Proteins

*S. cerevisiae* CMG (Cdc45/GINS/Mcm2-7) was overexpressed in yeast and purified as previously described (*Georgescu et al., 2014*). *E. coli* DnaB was overexpressed in *E. coli* and purification was as previously described (*Yuzhakov et al., 1996*). RPA was overexpressed in *E. coli* and purification was as previously described (*Henricksen et al., 1994*).

### Helicase assay substrates

For all radiolabeled oligos, 10 pmols of oligo were labeled at the 5' terminus with 0.05 mCi [$\gamma$-$^{32}$P]-ATP using T4 Polynucleotide Kinase (New England Biolabs) in a 25 µl reaction for 30' @ 37°C according to the manufacturer's instructions. The kinase was heat inactivated for 20' at 80°C. For annealing, 4 pmols of the radiolabeled strand were mixed with 6 pmols of the unlabeled complementary strand, NaCl was added to a final concentration of 200 mM, and the mixture was heated to 90°C and cooled slowly to room temperature. DNA oligos used in this study are listed in *Table 1*.

### Assay for CMG tracking over a flush duplex DNA

The DNA substrate in *Figure 2* contained three oligos annealed together as described above. The oligos used were: Paired duplex LEAD + 3' tail, 50 duplex LAG, and 5'-$^{32}$P-flush duplex LAG (see *Table 1*). Reactions contained 40 nM CMG, 0.5 nM DNA substrate and 1 mM ATP in 45 µl final volume of buffer A (20 mM Tris Acetate pH 7.6, 5 mM DTT, 0.1 mM EDTA, 10 mM MgSO$_4$, 30 mM KCl, 40 µg/ml BSA). Reactions were mixed on ice and started by placing in a water bath at 30°. At the indicated times, 12 µl aliquots were removed, stopped with buffer containing 20 mM EDTA and 0.1% SDS (final concentrations), and flash frozen in liquid nitrogen to prevent any unwanted reannealing that could possibly occur. Frozen reaction products were thawed quickly in lukewarm water and separated on 15% Native PAGE minigels. Gels were washed in distilled water, mounted on Whatman 3 MM paper, wrapped in plastic and exposed to a storage phosphor screen that was scanned on a Typhoon 9400 laser imager (GE Healthcare). Scanned gels were analyzed using Image-Quant TL v2005 software to obtain the quantitation shown in *Figure 2*. For all quantitations of helicase assays, the small background % of unannealed radiolabeled primer in the 'No CMG' lane was subtracted from the % unwound at each time point.

### Helicase assays using a dual biotin fork DNA

The forked DNAs in *Figure 3* contained a dual biotinylated oligo and a 5' $^{32}$P label on the oligo that was not biotinylated. The sets of two oligos used for the experiments were: (1) Forked DNA with biotins in the tracking strand (*Figure 3A*, left): 50 duplex LEAD dual biotin and 5'-$^{32}$P-50 duplex LAG (*Table 1*). The biotins were 12 and 25 nucleotides from the forked junction. (2) Forked DNA with biotins in the nontracking strand (*Figure 3A*, right): 50 duplex LAG dual biotin and 5'-$^{32}$P-50duplex LEAD (*Table 1*). The biotinylated nucleotides are 13 and 20 bases from the forked junction. Oligos were annealed as described above. Reactions were in a final volume of 45 µl Buffer A with 1 mM ATP; 0.5 nM forked dual biotinylated DNA was preincubated with or without 4 µg/ml streptavidin at 30°C for 5' and then on ice for 10', 20 nM CMG was added, and reactions were started by placing in a water bath at 30°. At the indicated times, 12 µl aliquots were removed, stopped with buffer containing 20 mM EDTA and 0.1% SDS (final concentrations), and flash frozen in

liquid nitrogen to prevent possible re-annealing. Reaction products were thawed quickly in lukewarm water and separated on 10% Native PAGE minigels and analyzed by phosphoimagery as above.

For the experiments of *Figure 3B*, reactions were mixed on ice and contained 0.5 nM radiolabeled DNA substrate and 100 nM *E. coli* DnaB in a buffer containing 20 mM Tris Acetate pH 7.6, 5 mM DTT, 0.1 mM EDTA, 10 mM MgSO$_4$, 50 mM potassium glutamate, 40 µg/ml BSA and 5 mM ATP in a total reaction volume of 45 µl. Reactions were started by incubating at 37°C and 12 µl aliquots were removed, stopped with EDTA/SDS, and flash frozen in liquid nitrogen at the time points indicated in the Figure. Because DnaB translocates 5'−3' on ssDNA, the tracking strand and nontracking strand substrates were the reverse of those used with CMG in *Figure 3A*.

## Helicase assays using a single biotinylated fork DNA

The forked DNAs in *Figure 4A, B* contained a single biotinylated oligo and a 5' $^{32}$P on the oligo that was not biotinylated. The sets of two oligos used for the experiments were: (1) Forked DNA with a biotinylated nucleotide in the tracking strand (*Figure 4A*): 50 duplex LEAD single biotin and 5'-$^{32}$P-50 duplex LAG (*Table 1*). The biotin is 12 nucleotides from the forked junction. (2) Forked DNA with a biotin in the nontracking strand (*Figure 4B*): 50 duplex LAG single biotin and 5'-$^{32}$P-50 duplex LEAD (*Table 1*). The biotinlyated nucleotide is 13 bases from the forked junction. Oligos were annealed as described above. Reactions were in a final volume of 45 µl Buffer A with 1 mM ATP; 0.5 nM forked dual biotinylated DNA was preincubated with or without 2 µg/ml streptavidin at 30°C for 5' and then on ice for 10', 20 nM CMG was added, and reactions were started by placing in a water bath at 30°. At the indicated times, 12 µl aliquots were removed, stopped with buffer containing 20 mM EDTA and 0.1% SDS (final concentrations), and flash frozen in liquid nitrogen to prevent possible re-annealing. Reaction products were thawed quickly in lukewarm water and separated on 10% Native PAGE minigels and analyzed by phosphoimagery as above. (3) For the reaction in *Figure 4C*, 50 duplex LEAD single biotin was annealed to 5'$^{32}$P-50 duplex LAG single biotin. The biotins are 12 (Lead) and 13 (Lag) nucleotides from the forked junction. Reaction conditions were the same as for the single biotin fork experiments except the substrate was pre-incubated with or without 4 µg/ml streptavidin before addition of CMG.

## Helicase assays showing bypass or displacement of streptavidin from single biotinylated fork DNA

In *Figures 5* and *6*, the forked DNAs contained a 5' $^{32}$P on the single biotinylated oligo annealed to the unlabeled, non-biotinylated oligo. Reactions were otherwise the same as in *Figure 4*, but an additional reaction was performed in which, after the substrate was pre-incubated with 2 µg/ml streptavidin, free D-biotin was added to a final concentration of 750 nM to prevent re-binding of any streptavidin displaced by CMG during the unwinding reaction.

## Helicase assays using a single biotinylated lagging strand fork DNA with or without RPA

For *Figure 6B*, the experiment in *Figure 6A* was repeated under conditions that prevent CMG from loading onto unwound ssDNA and removing streptavidin from biotin after unwinding the duplex. We previously showed that RPA completely eliminates CMG loading when added before CMG but not when CMG is pre-incubated with the substrate (in the absence of nucleotide) before addition of RPA (*Georgescu et al., 2014*). Accordingly, we loaded CMG in the absence of ATP by pre-incubating the substrate for 10' at 30°C with 45 nM CMG and 2 µg/ml streptavidin and started the reaction by addition of ATP (see reaction scheme at the top of *Figure 6B*). 1' before adding ATP, free biotin was added (750 nM final concentration) as a trap for displaced streptavidin and 1' after adding ATP, RPA was added to a final concentration of 25 nM to prevent further binding of CMG to DNA.

## Helicase assays using a substrate with M.HpaII covalently bound to the lagging strand in the duplex

For the assays of *Figure 7*, unlabeled M.HpaII LAG was annealed to 5'-$^{32}$P-M.HpaII LEAD (oligo sequences in *Table 1*). The fork duplex contains the recognition site for the M.HpaII methylase (CCGG), and the second dC position in the recognition site on the lagging strand is replaced with a 5-fluorodeoxycytidine to trap a covalent intermediate between the DNA substrate and the enzyme

during the methylation reaction (*Chen et al., 1991*). Covalent modification of the substrate was performed as previously described (*Duxin et al., 2014*), and the modified substrate was used at 0.5 nM in a standard unwinding reaction containing 1 mM ATP and 20 nM (*Figure 7A*) or 40 nM (*Figure 7B*) CMG.

## Additional information

### Funding

| Funder | Grant reference number | Author |
| --- | --- | --- |
| Howard Hughes Medical Institute | | Lance Langston<br>Mike O'Donnell |
| National Institutes of Health | GM38839 | Mike O'Donnell |

The funders had no role in study design, data collection and interpretation, or the decision to submit the work for publication.

### Author contributions

LL, Conceptualization, Investigation, Writing—original draft, Writing—review and editing; MO'D, Conceptualization, Writing—original draft, Writing—review and editing

### Author ORCIDs

Lance Langston, http://orcid.org/0000-0002-2736-9284
Mike O'Donnell, http://orcid.org/0000-0001-9002-4214

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
