## [Decision Letter]

Thank you for submitting your article "Both DNA Strands Enter the Central Channel of Eukaryotic CMG Helicase During Unwinding" for consideration by *eLife*. Your article has been reviewed by three peer reviewers and the evaluation has been overseen by a Reviewing Editor and Michael Marletta as the Senior Editor.

The reviewers have discussed the reviews with one another and the Reviewing Editor has drafted this decision to help you prepare a revised submission.

In this work, Langston & O'Donnell study the strand separation mechanism of the yeast CMG complex, comprising the replicative helicase MCM2-7, the GINS complex, and Cdc45. In particular, they address whether the *S. cerevisiae* version of this complex unwinds DNA by exclusion of the lagging DNA strand from the CMG central channel, as is commonly believed. The authors demonstrate that the CMG complex can translocate over double-stranded DNA (dsDNA). Using biotin-streptavidin blocks on either the leading (tracking) strand or the lagging (displaced) strand, they show that a streptavidin block on the displaced strand can either block the helicase, be bypassed, or be removed by the helicase. These findings suggest that the displaced strand travels through, or at least partly through, the central channel together with the tracking strand. In general, the authors argue their case well and the conclusions, if correct, are important.

However, at this point, the data do not unambiguously support the conclusion that "both strands enter the central channel during unwinding." There was much discussion, and a number of detailed comments were raised. In general, the authors make the assumption that just because biotin-streptavidin attached to each strand inhibits translocation that both strands pass through the central channel. It is also possible that one strand passes through the central channel and the other is engaged by the exterior of the Mcm2-7 ring or by Cdc45 or GINS, and acting in a regulatory capacity as has been shown or proposed for DnaB and phage replicative helicases. Such interactions could be important for DNA unwinding and could impact DNA unwinding while not passing through the central channel. The authors do, however, show that the same DNA constructs have the canonical effect on DnaB (i.e., it is not blocked by an obstruction on the displaced strand). But these and other findings are interpreted by the authors in only one way, namely, that the displaced strand runs, at least partially, through the central cavity of CMG, which is in seeming contradiction with a large body of evidence in the field. The entire conclusion of this paper rests on the interpretation of a single experimental protocol, namely, the effects of streptavidin-biotin conjugates on DNA unwinding.

However, those data, though generally nicely controlled and executed, are themselves inconsistent. Specifically, even though single streptavidin-biotin conjugates are bypassed on either strand, the streptavidin is displaced from the leading strand but mostly not (only 10% at early times) from the displaced strand. This result is the finding expected if the displaced strand did not pass through the central cavity, nor was subjected to any form of mechanical stripping. However, when two streptavidin-biotin conjugates are used on one strand, the behavior changes, and streptavidin-biotin conjugates on either strand block unwinding. This result with the double streptavidin-biotin conjugate is peculiar, and why or how two streptavidin-biotin conjugates block unwinding is not explained adequately, but this finding is the major reason for proposing that the displaced stand goes through CMG. So, the data can support either model. Furthermore, for both single- and double-biotin attachments, inhibition on the lagging-strand template is consistently less than that observed on the leading-strand template and only leading-strand templates that have had streptavidin removed can be displaced but 75-90% of the displaced lagging-strand templates retain streptavidin. Put another way, these results show that there is no obligatory requirement for displacement of the streptavidin from the lagging strand, upon DNA unwinding – so, does the dsDNA *not* enter the channel 75-90% of the time? This latter finding indicates that a lagging strand template still bound to streptavidin can be displaced, arguing strongly against the authors’ hypothesis that the lagging strand passes through the central channel.

Nonetheless, given the provocative nature of the findings and the high quality of the work, the authors should be given the opportunity to address these comments in a revised version that would be reconsidered. To condense an extensive discussion by the reviewers and editor into two major comments, that authors need to address the following (though seemingly extensive, the first comment is a request for an additional essential experiment, and the second comment requires only revisions of the text):

1) As mentioned above, the results obtained with the single-streptavidin-biotin displacement experiment contradict the results with the double streptavidin-biotin experiments. The former support a more conventional model for CMG, whereas the latter support the authors' model. This issue must be resolved experimentally. Rather than using a non-covalent streptavidin-biotin conjugate, a covalent block would be better. Placing a bulky covalent adduct on the lagging strand would resolve the ambiguity of whether or not that strand passes through or around CMG, without being compromised by whether or how streptavidin is bound. This result should be unequivocal and would not need the additional controls with streptavidin-biotin. This approach was used successful with RecBCD helicase, a helicase through which ssDNA passes inside central channels (see, e.g., Dziegielewska et al. J Mol Biol. 2006;361(5):898-919. doi: 10.1016/j.jmb.2006.06.068. PMID: 16887143). Among the many covalent adducts that could be made, the authors should consider one or more of the following. The simplest is to synthesize DNA with a large adduct (e.g., a fluorescent dye) in a precise location; such fluorescently-modified oligos are commonplace, and the dyes are ~1,000 daltons which may be large enough, but structural modeling might be informative. The second alternative is a method used by Walter and colleagues in which a fluorinated cytosine in the context of an HpaII methylation site is present in the DNA. Addition of HpaII to this creates a covalent adduct between HpaII and the DNA (Duxin, et al. (2014). Cell, 159(2), 346-357. http://doi.org/10.1016/j.cell.2014.09.024). Importantly, this has been used to show that the adduct stalls eukaryotic replication forks in the paper above. A third alternative method of forming a covalent bulky adduct is to use an amino-modifier-C6-dT (IDT, Sigma) and then crosslink this to any lysine containing protein using a divalent NHS-ester such as BS3, DSG, or DST (11.4, 7.7, 6.4 Å linker length, available from Pierce). This way, a range of protein sizes and linker lengths can be tested. Regardless of the method, an experiment with a bulky covalent adduct on the lagging strand is essential.

2) The authors need to tone down their conclusions, and be less dogmatic about the possibility that the displaced strand travels through the central channel. A more balanced and measured presentation that considers multiple possible explanations and acknowledges that there are aspects of the data that are not consistent with a dsDNA through the central channel model (Figure 6 being the most notable) is essential (i.e., limiting the speculation to the Discussion, rather than highlighting it throughout the manuscript: the Title, Abstract, and Methods). In this regard, the authors need to discuss and evaluate more completely:

a) In terms of structural evidence, there is a 6-8 Å resolution cryo-EM structure of a fork bound to the CMG which shows ssDNA in the central channel. This structure needs to be discussed by the authors.

b) In the Discussion, the authors describe a possible exit channel from the CMG helicase for the displaced strand based on their previous cryo-EM structure of the same complex (Discussion, seventh paragraph). It would be helpful if they would show this path in the cryo-EM structure. This figure could replace Figure 7, which contains little added information as it is very similar to Figure 1.

c) There is also data that contradicts their findings using complete replication forks in *Xenopus* extracts. Although the data do not exclude the possibility that dsDNA passes through the central channel, they should be discussed more completely.

d) Another model that could explain the model is that the lagging-strand template does not pass through the central channel but has strong interactions with Cdc45/GINS and that these interactions are important for DNA unwinding. For example, in several of the CMG structures there is enough space for a ssDNA to pass between the Cdc45/GINS proteins and the Mcm2-7 ring.

e) Definitive proof can only come from a structure of CMG with a forked DNA bound, but this clearly lays beyond the scope of this work. Are the authors aware of any work in press that might provide structural insight?

f) In light of the mixed behavior it seems when CMG helicase encounters a block on the displaced strand (i.e., it can either stop at, bypass, or displace the streptavidin; Figure 6), it will be informative to discuss the recent results of Elshenawy et al. (Nature 2015, vol 525, pp394-398) who show a similar mixed behavior for the *E. coli* DnaB, where depending on the speed of the replisome, the helicase can either displace the Tus protein or be stopped in its tracks.

---

## [Author Response]

*[…] However, at this point, the data do not unambiguously support the conclusion that "both strands enter the central channel during unwinding." There was much discussion, and a number of detailed comments were raised. In general, the authors make the assumption that just because biotin-streptavidin attached to each strand inhibits translocation that both strands pass through the central channel. It is also possible that one strand passes through the central channel and the other is engaged by the exterior of the Mcm2-7 ring or by Cdc45 or GINS, and acting in a regulatory capacity as has been shown or proposed for DnaB and phage replicative helicases. Such interactions could be important for DNA unwinding and could impact DNA unwinding while not passing through the central channel. The authors do, however, show that the same DNA constructs have the canonical effect on DnaB (i.e., it is not blocked by an obstruction on the displaced strand). But these and other findings are interpreted by the authors in only one way, namely, that the displaced strand runs, at least partially, through the central cavity of CMG, which is in seeming contradiction with a large body of evidence in the field. The entire conclusion of this paper rests on the interpretation of a single experimental protocol, namely, the effects of streptavidin-biotin conjugates on DNA unwinding.*

We now add the information on our recently published structure of CMG bound to a replication fork, showing dsDNA entry into CMG. Even with dsDNA entering CMG, which nicely explains the data in the submitted paper, the overall conclusion that we describe is a modified steric exclusion process, in which this process has an internal unwinding point, contrary to the classic steric exclusion model, but consistent with the CMG-forked DNA structure and with the data of this report.

*However, those data, though generally nicely controlled and executed, are themselves inconsistent. Specifically, even though single streptavidin-biotin conjugates are bypassed on either strand, the streptavidin is displaced from the leading strand but mostly not (only 10% at early times) from the displaced strand. This result is the finding expected if the displaced strand did not pass through the central cavity, nor was subjected to any form of mechanical stripping. However, when two streptavidin-biotin conjugates are used on one strand, the behavior changes, and streptavidin-biotin conjugates on either strand block unwinding. This result with the double streptavidin-biotin conjugate is peculiar, and why or how two streptavidin-biotin conjugates block unwinding is not explained adequately, but this finding is the major reason for proposing that the displaced stand goes through CMG. So, the data can support either model. Furthermore, for both single- and double-biotin attachments, inhibition on the lagging-strand template is consistently less than that observed on the leading-strand template and only leading-strand templates that have had streptavidin removed can be displaced but 75-90% of the displaced lagging-strand templates retain streptavidin. Put another way, these results show that there is no obligatory requirement for displacement of the streptavidin from the lagging strand, upon DNA unwinding – so, does the dsDNA not enter the channel 75-90% of the time? This latter finding indicates that a lagging strand template still bound to streptavidin can be displaced, arguing strongly against the authors’ hypothesis that the lagging strand passes through the central channel.*

All the data is consistent with one model, a modified steric exclusion model, and this is explained in the revised manuscript.

*Nonetheless, given the provocative nature of the findings and the high quality of the work, the authors should be given the opportunity to address these comments in a revised version that would be reconsidered. To condense an extensive discussion by the reviewers and editor into two major comments, that authors need to address the following (though seemingly extensive, the first comment is a request for an additional essential experiment, and the second comment requires only revisions of the text):*

*1) As mentioned above, the results obtained with the single-streptavidin-biotin displacement experiment contradict the results with the double streptavidin-biotin experiments. The former support a more conventional model for CMG, whereas the latter support the authors' model. This issue must be resolved experimentally. Rather than using a non-covalent streptavidin-biotin conjugate, a covalent block would be better. Placing a bulky covalent adduct on the lagging strand would resolve the ambiguity of whether or not that strand passes through or around CMG, without being compromised by whether or how streptavidin is bound. This result should be unequivocal and would not need the additional controls with streptavidin-biotin. This approach was used successful with RecBCD helicase, a helicase through which ssDNA passes inside central channels (see, e.g., Dziegielewska et al. J Mol Biol. 2006;361(5):898-919. doi: 10.1016/j.jmb.2006.06.068. PMID: 16887143). Among the many covalent adducts that could be made, the authors should consider one or more of the following. The simplest is to synthesize DNA with a large adduct (e.g., a fluorescent dye) in a precise location; such fluorescently-modified oligos are commonplace, and the dyes are ~1,000 daltons which may be large enough, but structural modeling might be informative. The second alternative is a method used by Walter and colleagues in which a fluorinated cytosine in the context of an HpaII methylation site is present in the DNA. Addition of HpaII to this creates a covalent adduct between HpaII and the DNA (Duxin, et al. (2014). Cell, 159(2), 346-357. http://doi.org/10.1016/j.cell.2014.09.024). Importantly, this has been used to show that the adduct stalls eukaryotic replication forks in the paper above. A third alternative method of forming a covalent bulky adduct is to use an amino-modifier-C6-dT (IDT, Sigma) and then crosslink this to any lysine containing protein using a divalent NHS-ester such as BS3, DSG, or DST (11.4, 7.7, 6.4 Å linker length, available from Pierce). This way, a range of protein sizes and linker lengths can be tested. Regardless of the method, an experiment with a bulky covalent adduct on the lagging strand is essential.*

We have synthesized a covalent protein adduct on the lagging strand using one of the suggestions given to us, and have performed the additional experiment that the reviewers have requested. We felt the fluorophore was too small, especially considering the Johannes Walter data in the paper that reveals that CMG can go over a small peptide attached to DNA. We also note that a 2014 JBC paper from Hiroshi Ide’s lab (PMID: 23283980) used a variety of different proteins covalently linked to DNA through an oxanine in the duplex (their invention) and observed that proteins of < 5 kDa did not inhibit DnaB, T7gp4, Mcm467 or SV40 T-Antigen when attached to the tracking strand so we felt that the fluorescent dye would not serve the desired purpose. Thus, we tried both the NHS ester and M.HpaII covalent DNA approaches. We spent considerable time and money doing these, and finally succeeded in obtaining the M.HpaII covalent adduct. The NHS ester attempts were all failures and we don’t know why.

Regardless, the M.HpaII adduct was the most desirable method, considering that it has already been used in the *Xenopus* system to show that the lagging-HpaII adduct was bypassed in *Xenopus* extracts as noted by the reviewers. The results of the M.HpaII-lagging strand adduct showed that it was a complete block to isolated pure CMG helicase. Heat denaturation of the covalent HpaII gave a low, but measurable bypass. This experiment is now added as Figure 7 to the revised manuscript as requested by the reviewers. Presumably another factor within the *Xenopus* extract facilitates CMG bypass of a covalent lagging strand M.HpaII block.

*2) The authors need to tone down their conclusions, and be less dogmatic about the possibility that the displaced strand travels through the central channel. A more balanced and measured presentation that considers multiple possible explanations and acknowledges that there are aspects of the data that are not consistent with a dsDNA through the central channel model (Figure 6 being the most notable) is essential (i.e., limiting the speculation to the Discussion, rather than highlighting it throughout the manuscript: the Title, Abstract, and Methods). In this regard, the authors need to discuss and evaluate more completely:*

*a) In terms of structural evidence, there is a 6-8 Å resolution cryo-EM structure of a fork bound to the CMG which shows ssDNA in the central channel. This structure needs to be discussed by the authors.*

We appreciate this comment, and have discussed a variety of non-dogmatic explanations in the Discussion. We did cite the Costa 6-8 Å work that contains 6 nucleotides inside CMG in the original, and revised paper. In fact, we have now published yeast CMG-ssDNA at 4.8 Å that has 14 bases of ssDNA inside the channel. However, in both cases non-hydrolyzable nucleotide was used and this might explain why neither structure observed CMG at the forked junction, and therefore there is not much one can say from those structures about how CMG engages the fork and deals with blocks.

Importantly, we succeeded in obtaining a CMG-forked DNA structure (using ATP) just when this blocking paper was finished. The CMG-forked DNA structure work is now published, and it is very informative to this blocking study (and vice versa). As the reviewers requested, we discuss the structural evidence, in detail in the revised manuscript. Considering the CMG-forked DNA has several unique features not seen in other helicases, along with the fact that the blocking experiments test facets of the structure, we show a summary of the structure in Figure 1 for the convenience of the reader since it is central to understanding and interpreting the results of the paper.

The CMG-forked DNA structure shows dsDNA entry into the CMG, and an internal unwinding point, features that are not consistent with a classic steric exclusion model, and which are consistent with the blocking observations of this report. But the dsDNA does not penetrate too far into CMG (i.e. just past the zinc fingers), and a side channel is likely not needed since the dsDNA is not too far inside. The lagging strand is not visible (i.e. probably too mobile or located in multiple locations) so whether there is a side channel is not rigorously resolved. We take the viewpoint here that CMG works by a modified steric exclusion process, in which CMG binds dsDNA and has an internal unwinding point (like the side channel extrusion model), but since the lagging strand is very close to the surface, it just goes back out the same channel and is thereafter excluded from the channel. This interpretation brings the structure and blocking data in congruence to Johannes Walter’s work in *Xenopus* extracts, in which the CMG is concluded to function by steric exclusion.

Importantly, there is still room for many different interpretations of the data, and we are compelled by the data not to be dogmatic considering the structure was not definitive as to the disposition of the lagging strand. Alternative ways of explaining the results are discussed, and the conclusions are stated as speculative and with alternatives.

*b) In the Discussion, the authors describe a possible exit channel from the CMG helicase for the displaced strand based on their previous cryo-EM structure of the same complex (Discussion, seventh paragraph). It would be helpful if they would show this path in the cryo-EM structure. This figure could replace Figure 7, which contains little added information as it is very similar to Figure 1.*

The side channel in an inactive MCM double hexamer is not of much relevance since it is not an active helicase, and the proposed side channel is much further into the CMG than the dsDNA in the CMG-fork structure. Rather than reproducing the figure of a MCM double hexamer side channel from the published work, we now specify the exact figure in the original paper along with the citation to that work, making it easy for an interested reader to find the information. We have replaced Figure 7 in the revised manuscript with the figure discussed in the comment above.

*c) There is also data that contradicts their findings using complete replication forks in Xenopus extracts. Although the data do not exclude the possibility that dsDNA passes through the central channel, they should be discussed more completely.*

The data from *Xenopus* extracts does not contradict the work of this report, nor does our recent structural finding of dsDNA entry into the central channel of yeast CMG contradict the *Xenopus* work – as discussed in the revised manuscript. Our conclusion of CMG unwinding is by a modified steric exclusion mechanism, as stated above, and consistent with the *Xenopus* work. The isolated CMG circumvents a single streptavidin block, usually without displacing streptavidin, but it is slowed 50%. It is slowed further if there is a dual biotin-SA block (in additive fashion). Please note that the *Xenopus* study did not test whether any of the lagging SA is displaced upon fork passage, so contradictions on this aspect can’t be evaluated. The elegant work of the Walter lab did add free biotin to the extract and concluded that this did not affect their results (Supplemental Figure 3), but they didn’t examine whether the SA was displaced from or retained on the DNA. We also note that free biotin did not affect our results either and it wouldn’t be expected to do so. If SA is displaced from the DNA, it doesn’t matter much whether it rebinds after the helicase has bypassed the block, so that’s not a useful control. The only reason to use a free biotin trap is to determine whether SA is displaced from the DNA or retained on the DNA as we did here. In this way, we reveal features of the reaction that were not examined (and could not be examined) in the *Xenopus* extract work. The only difference between the two studies is the efficiency of bypass. Using a *Xenopus* extract, the streptavidin data shows a difference +/- SA at the first 10 min point, but there is no quantitation of the gel, while in our work 10 min is the last time point. The same *Xenopus* extract study also performs single-molecule work using a Dig-block and Dig antibody coupled to a quantum dot. That work is quantitated (Figure 4), and is reported to give a 20-26% arrest (their term) of the fork. This is within 2 fold of our observed 50% slow down with a single biotin-SA on the lagging strand. But overall – please note that we do not dispute that an extract would be more efficient in bypass compared to isolated CMG – considering the many proteins in an extract. We discuss and cite the *Xenopus* extract study in the Introduction, Discussion and throughout. We propose that an additional factor(s) might be present in the extract that enables more efficient bypass of pure isolated CMG.

*d) Another model that could explain the model is that the lagging-strand template does not pass through the central channel but has strong interactions with Cdc45/GINS and that these interactions are important for DNA unwinding. For example, in several of the CMG structures there is enough space for a ssDNA to pass between the Cdc45/GINS proteins and the Mcm2-7 ring.*

We have thought of this possibility but the high resolution structure of yeast CMG (PMID 27310307), at 3.8 A, shows that the secondary channel is completely filled by side chains at high resolution. There isn’t room for ssDNA or anything, at least in the yeast CMG. We have now stated this in the revised manuscript, and cited the high resolution cryoEM study. That said, other CMGs may be different.

*e) Definitive proof can only come from a structure of CMG with a forked DNA bound, but this clearly lays beyond the scope of this work. Are the authors aware of any work in press that might provide structural insight?*

Figure 1 now includes the CMG-forked DNA structure.

*f) In light of the mixed behavior it seems when CMG helicase encounters a block on the displaced strand (i.e., it can either stop at, bypass, or displace the streptavidin; Figure 6), it will be informative to discuss the recent results of Elshenawy et al. (Nature 2015, vol 525, pp394-398) who show a similar mixed behavior for the E. coli DnaB, where depending on the speed of the replisome, the helicase can either displace the Tus protein or be stopped in its tracks.*

We find the results of Elshenawy et al. (Nature 2015, vol 525, pp394-398) very interesting. However, we feel it is too early to explain the results of this report in terms of replisomes (helicases) that move at different intrinsic speeds. Single molecule studies are in progress on this work, and if we notice a correlation between streptavidin displacement/bypass with helicase speed, we will certainly cite the Eishenaway et al. study in that manuscript.